# On the impact of true polar wander on heat flux patterns at the core-mantle boundary

Thomas Frasson[1], Stéphane Labrosse[2], Henri-Claude Nataf[1], Nicolas Coltice[3], and Nicolas Flament[4]

[1]Univ. Grenoble Alpes, Univ. Savoie Mont Blanc, CNRS, IRD, Univ. Gustave Eiffel, ISTerre, 38000 Grenoble, France
[2]ENS de Lyon, Université Lyon-1, LGL-TPE, 46 allée d'Italie, 69007 Lyon.
[3]Université Côte d'Azur, CNRS, Observatoire de la Côte d'Azur, IRD, GEOAZUR, France
[4]Environmental Futures, School of Earth, Atmospheric and Life Sciences, University of Wollongong, Northfields Avenue, NSW 2522, Australia

**Correspondence:** Thomas Frasson (thomas.frasson@univ-grenoble-alpes.fr)

**Abstract.** The heat flux across the core-mantle boundary (CMB) is a fundamental variable for Earth evolution and internal dynamics. Seismic tomography provides access to seismic heterogeneities in the lower mantle, which can be related to present-day thermal heterogeneities. Alternatively, mantle convection models can be used to either infer past CMB heat flux or to produce statistically realistic CMB heat flux patterns in self-consistent models. Mantle dynamics modifies the inertia tensor of the Earth, which implies a rotation of the Earth with respect to its spin axis, a phenomenon called true polar wander (TPW). This rotation must be taken into account to link the dynamics of the mantle to the dynamics of the core. In this study, we explore the impact of TPW on the CMB heat flux over long timescales ($\sim$ 1 Gyr) using two recently published mantle convection models: one model driven by a plate reconstruction and a second that self-consistently produces a plate-like behaviour. We compute the geoid in both models to correct for TPW. In the plate-driven model, we compute a total geoid and a geoid in which lateral variations of viscosity and density are suppressed above 350 km depth. An alternative to TPW correction is used for the plate-driven model by simply repositioning the model in the original paleomagnetic reference frame of the plate reconstruction. The average TPW rates range between 0.4° Myr$^{-1}$ and 1.8° Myr$^{-1}$, but peaks up to 10° Myr$^{-1}$ are observed. We find that in the plate-driven mantle convection model used in this study, the maximum inertia axis produced by the model does not show a long-term consistency with the position of the magnetic dipole inferred from paleomagnetism. TPW plays an important role in redistributing the CMB heat flux, notably at short time scales ($\leq$ 10 Myr). Those rapid variations modify the latitudinal distribution of the CMB heat flux, which is known to affect the stability of the magnetic dipole in geodynamo simulations. A principal component analysis (PCA) is computed to obtain the dominant CMB heat flux pattern in the different cases. These heat flux patterns are representative of the mantle convection cases studied here and can be used as boundary conditions for geodynamo models.

## 1 Introduction

Temperature heterogeneities in the lower mantle impose a heterogeneous heat flux at the top of the core, across the core-mantle boundary (CMB). This CMB heat flux is an important variable of Earth's thermal evolution and dynamics, especially

for core convection and the geodynamo. The mean CMB heat flux controls the core cooling rate, which determines the power available for the geodynamo. Both the CMB heat flux mean value and lateral variations affect dynamo behaviour in numerical simulations, with strong effects on magnetic reversal frequency and on the angle between spin and magnetic dipole axes (Glatzmaier et al., 1999; Kutzner and Christensen, 2004; Olson et al., 2010). Large heat flux heterogeneities can even prevent dynamo action (Olson and Christensen, 2002). It is therefore important to evaluate what could be the evolution of the CMB heat flux on geologic timescales, in order to assess consequences for the geodynamo.

Because viscosity is much larger in Earth's mantle than in Earth's outer core, the CMB is an isothermal boundary for the mantle, while the core sees the CMB as an imposed laterally varying heat flux. This heat flux changes on mantle convection timescales, which are much larger than core dynamics timescales. Our understanding of the CMB heat flux, and notably its spatial distribution, depends on our knowledge of lower mantle structure and dynamics. Seismic tomography offers a view of the lowermost mantle, revealing more and more complex structures (e.g. Dziewonski et al., 1977; Lay and Helmberger, 1983; Garnero and Helmberger, 1995; Su and Dziewonski, 1997; Durand et al., 2017, see Ritsema and Lekić 2020 for a review). Around the equator, two antipodal large low velocity provinces (LLVPs) are particularly conspicuous, and are interpreted as thermochemical piles (Garnero and McNamara, 2008). They form a characteristic structure dominated by spherical harmonics degree 2. LLVPs are correlated with the degree 2 geoid, positive geoid anomalies being observed over each LLVP (Dziewonski et al., 1977). Recent works suggest a chemically distinct composition at the base of these structures, stabilizing them by imposing a negative buoyancy (Richards et al., 2023). Such stable structures in the lower mantle would act as thermal insulator for the core and have significant implications for the CMB heat flux (Nakagawa and Tackley, 2008; Liu and Zhong, 2015). At scales too small to be resolved in global tomographic models, ultra low velocity zones (ULVZ) have also been observed using dedicated approaches (e.g. Garnero and Helmberger, 1995; Rost et al., 2005). Despite these improvements, it is still difficult to have a clear view of the CMB heat flux pattern. Thermal and chemical effects are notably difficult to distinguish in tomographic models (Trampert, 2004; Mosca et al., 2012), which only provides a present-day snapshot of Earth's history.

Records of eruption sites of hotspots suggest that LLVPs could have remained fixed for the past 300 Myr at least (Burke et al., 2008; Torsvik et al., 2010; Dziewonski et al., 2010), providing a stable large-scale heat flux pattern through time. This view of stable LLVPs has however been challenged by recent seismic tomography models (Davaille and Romanowicz, 2020) and mantle flow reconstructions (Flament et al., 2022). Past CMB heat flux estimates have been obtained from reconstructions of mantle flow driven by observed plate motions for the past 450 Myr (Zhang and Zhong, 2011; Olson et al., 2015), and more recently for the past 1 Gyr (Flament et al., 2022). These models show that the CMB heat flux pattern is governed by plate motion through subducted slabs, which cool the lower mantle. Large lowermost mantle chemical piles stay warmer than the surrounding mantle, thereby increasing overall lateral heterogeneities of CMB heat flux. CMB heat flux reconstructions can thus be used to constrain core evolution and magnetic field generation. Olson et al. (2015) used a plate-driven mantle convection model to drive a thermal evolution model of the core, while Zhang and Zhong (2011) found equatorial heat flux minima around 170 Myr and 100 Myr ago, which coincide with Kiaman and Cretaceous magnetic superchrons, respectively.

These models are useful to explore the past of mantle convection. They are however limited by the accuracy of plate reconstructions which remain poorly constrained before the assembly of Pangea (Müller et al., 2022; Seton et al., 2023). Al-

ternatively, CMB heat flux estimates can be obtained from self-consistent models, without prescribed surface velocities (Liu and Zhong, 2015; Coltice et al., 2019). This approach is less "Earth-like" than models with prescribed plate motion in that it does not aim to reproduce Earth's actual past. However, it makes it possible to obtain statistically realistic information on mantle convection depending on input parameters. Using this kind of model, Nakagawa and Tackley (2008) notably showed that lateral variations in the CMB heat flux could be as large as the mean CMB heat flux.

Earth's spin plays a crucial role in core dynamics. If we are to explore the impact of realistic CMB heat flux patterns on the geodynamo, it is essential that these patterns be produced in a reference frame that preserves the spin axis. Mantle convection simulations do not depend on the position of Earth's spin axis since rotational forces are negligible in the mantle, and surface and CMB boundary conditions are not affected by a global rotation of the mantle with respect to its spin axis. However, mass redistribution and boundary topographies caused by convection modify the moments of inertia of the mantle (Munk and MacDonald, 1960; Phillips et al., 2009), which can be obtained from the degree 2 coefficients of the geoid (Schaber et al., 2009). The mantle therefore rotates in order to keep its axis of greatest inertia along the spin axis (Goldreich and Toomre, 1969). This is called true polar wander (TPW). This TPW is usually not considered in mantle convection models, which means that the z-axis of the reference frames of the models are not aligned with the Earth's spin axis. The TPW can however be obtained in the models, by computing the inertia tensor of the mantle. TPW emerging from mantle convection models has notably been studied to retrieve the past track of the spin axis (Steinberger and O'Connell, 1997; Schaber et al., 2009), or to investigate the link between supercontinents and TPW in mantle convection models (Zhong et al., 2007; Phillips et al., 2009). Using this computed TPW, it is possible to rotate the model outputs so that the maximum inertia axis stays aligned with a fixed spin axis. This method is the only way to reposition the model in the reference frame of the spin axis in self-consistent models. TPW can also be corrected using this method in plate-driven models. In this case however, the past locations of continents can also be used to constrain the position of the mantle relative to the spin axis. If the plate reconstruction is in a paleomagnetic reference frame, the latitudes of continents relative to the spin axis are fixed and no correction is required. Despite this advantage, plate reconstructions in a paleomagnetic reference frame can be problematic when used as boundary conditions in mantle convection models because they include a significant amount of net rotation (Müller et al., 2022). Plate-driven mantle convection models thus classically use plate reconstructions placed in a mantle reference frame following various methods (Coltice et al., 2017; Cao et al., 2021; Müller et al., 2022; Flament et al., 2022). In this framework, only the present-day continent distributions are consistent with the position of the spin axis, and the previous positions of the spin axis are lost.

This work aims at describing the CMB heat flux produced by two up-to-date mantle convection models in the reference frame relevant for core dynamics. One of these models is driven by a plate reconstruction (Müller et al., 2022), while the other one is entirely self-consistent and free to evolve (Coltice et al., 2019). For each model, we notably provide representative CMB heat flux maps that can be used in geodynamo simulations. These maps are obtained using a principal component analysis (PCA), which brings out dominant CMB heat flux patterns. We extract CMB heat flux and geoid from the two different mantle convection models, compute TPW from the degree two of the geoid, and rotate the simulation frame accordingly to position CMB heat flux maps in the reference frame relevant to core dynamics. For the plate-driven mantle convection model, an alternative correction is performed by rotating the model in the paleomagnetic reference frame of the plate reconstruction. This

alternative correction is similar to what was done by Dannberg et al. (2024), i.e. running the mantle convection model using the plate reconstruction of Merdith et al. (2021) in the paleomagnetic reference frame (PMAG case in Müller et al. (2022)).

Section 2 describes the methods used for the successive steps of the analysis. Results are presented in Sect. 3 and discussed in Sect. 4. We conclude in Sect. 5.

## 2 Methods

### 2.1 3D mantle convection models

Our study rests upon an analysis of two published 3D mantle convection simulations. Both models simulate mantle convection in a 3D spherical shell including tectonic plates at the surface, and chemical piles at the bottom. They however differ in the way plate-tectonics is handled. The first model (similar to case NNR of Müller et al. (2022)), named MF in the following, is driven by plates with structure and kinematics derived from geological observations. This model thus aims at reproducing the actual history of Earth's mantle convection, using constraints from surface kinematics and the position of subduction zones. The second model (Coltice et al., 2019), named MC in the following, produces a plate-like behaviour without imposing any surface kinematics. Plate-like structures and subduction zones are obtained in a fully self-consistent way using a pseudo-plastic rheology and a strongly temperature-dependent viscosity. This model does not aim to reproduce the actual past or future history of mantle convection. Instead, it provides an alternative convection history with statistically realistic parameters. The following paragraphs describe the characteristics of those models relevant for this study.

#### 2.1.1 MF model

Model MF was computed using the code *CitcomS* (Zhong et al., 2008; Bower et al., 2015) under the extended-Boussinesq approximation. The model is driven at the surface by imposing plate velocities, the positions of subduction zones, and the age of the oceanic lithosphere. These constraints are taken from the plate reconstruction of Merdith et al. (2021) expressed in the no-net-rotation reference frame as in Müller et al. (2022). This no-net-rotation reference frame only differs from the original paleomagnetic reference frame of the plate reconstruction by net global rotations of the surface. The model is initialized at 1.25 Gyr ago (Ga) with a 250-Myr warm-up phase during which tectonic velocities, subduction zone positions and lithospheric ages are derived from the tectonic reconstruction at 1 Ga. The initial condition at 1.25 Ga consisted of slabs inserted down to 1,000 km depth and of a 113 km-thick basal layer of material with excess density $\delta\rho_p = 95.2$ kg m$^{-3}$. The continents have a density deficit of $\delta\rho_c = 140$ kg m$^{-3}$ on average (Flament et al., 2014). After the warm-up phase, surface boundary conditions are updated in one-million-year increments with linear interpolation at each numerical time step. The model is similar to case NNR of Müller et al. (2022), with differences listed below. The Rayleigh number (based on the thickness of the mantle) is equal to $10^8$ with an internal heating rate of 30 TW. The excess density of the basal layer was $\delta\rho_p/\rho_{100} = 1.7\%$, where $\rho_{100} = 5546$ kg m$^{-3}$ is the average density in the bottom 100 km of the mantle in the PREM model (Dziewonski and Anderson, 1981). Phase changes are considered at depths of 410 km, 670 km and 2740 km, and assumed to occur over a 40 km depth-range. At

410 km depth, the density change is 3% of ambient mantle density and the Clapeyron slope is equal to 4 MPa K$^{-1}$ (Billen, 2008, and references therein). At 670 km depth, the density change is 7% of ambient mantle density and the Clapeyron slope is $-2$ MPa K$^{-1}$ (Billen, 2008, and references therein). At 2740 km depth, the density change is 1.1% of ambient mantle density and the Clapeyron slope is 12 MPa K$^{-1}$ (Nakagawa and Tackley, 2014).

This model predicts CMB heat flux based on a reconstruction of past positions of tectonic plates and associated mantle flow. The evolution of the thermochemical basal boundary layer is dictated by subducting slabs, which are introduced at the surface following the plate model. Heat flux patterns are thus directly related to the past 1 Gyr of mantle convection history, which depends on the plate tectonic reconstruction imposed as surface boundary condition. Whilst the past 200 Myr are well constrained notably from magnetic anomalies and hotspot tracks preserved in the oceanic crust, plate motions are more uncertain for earlier geological times. Advantages of model MF are that it generally matches the present-day structure of the mantle (Flament et al., 2022; Müller et al., 2022), and is designed to reconstruct Earth's mantle convection based on available constraints.

### 2.1.2 MC model

Model MC reproduces in total 1131 Myr of mantle evolution using the code *StagYY* in a 3-D Yin-Yang geometry (Tackley, 2008) and under the Boussinesq approximation. This model has been set up to reproduce Earth-like mantle convection features, with particular attention to plate-like behaviour. Coltice et al. (2019) presented results from this model focusing on plate-tectonics. Here, we use the same model to study the CMB heat flux. At the base of the mantle, chemical piles are modelled as a material denser than the surrounding mantle. Continents are modelled as a compositionally distinct material that is less dense and more viscous than the ambient mantle, with 200 km-thick interiors and 125 km-thick rims. The reference density $\rho_0$ and viscosity $\eta_0$ have dimensional values 4000 kg m$^{-3}$ and $10^{22}$ Pa s, respectively. The negative compositional density anomaly inside continents is $\delta\rho_c = -225$ kg m$^{-3}$, and the compositional density excess in basal piles is $\delta\rho_p = 137$ kg m$^{-3}$. The dimensional internal heating rate is 33 TW.

The initial state of the simulation is an equilibrated mantle circulation obtained with two fixed antipodal 500 km-thick chemical piles around the equator, and fixed continents assembled in a Pangea-like supercontinent placed above the "Atlantic" pile. At the start of the simulation, continents and piles are allowed to move freely. The relaxation to a new statistically steady state takes about 300 Myr. The first 300 Myr are thus not considered in the following analysis.

In contrast with model MF, plate kinematics are not imposed in this model. A plate-like behaviour is self-consistently produced using a pseudo-plastic rheology and a temperature-dependent viscosity (Tackley, 2000a, b). An uppermost 14 km-thick weak layer in oceanic regions makes it possible to obtain asymmetric subduction zones (Crameri et al., 2012). Model MC reproduces a statistically realistic mantle convection that fits the observations of global features (plate dimensions and velocities, surface heat flux, hypsometry, plume buoyancy flux) as well as local features (continental breakup, rifting, back-arc extension, mantle plumes) as shown in Coltice et al. (2019). The extreme temperature-dependence of viscosity in the model gives rise to plumes that display kinematic, thermal and buoyancy properties similar to Earth's plumes (Arnould et al., 2020).

Regarding the objective of this work, the main advantages of model MC are the Earth-likeness of surface processes and its long time evolution. It captures the effect of realistic plate tectonics on a CMB heat flux that varies in space and time over nearly 1 Gyr. Model MC notably contains a complete cycle of breakup and assembly of a super-continent, which is thought to modulate CMB heat flux (Olson et al., 2013; Amit et al., 2015). The similarity between model MC and the Earth at the bottom of the mantle is less certain. Nevertheless, the presence of chemical piles allows for strong, large-scale temperature heterogeneities, as revealed by seismic tomography (Trampert, 2004; Mosca et al., 2012).

## 2.2 Geoid

TPW is controlled by the change in the Earth's moment of inertia around its spin axis. The moment of inertia is obtained from the degree 2 components of the geoid, which we thus need to compute for our mantle flow models. The geoid is the equipotential surface of gravity measured or computed around a reference level, sea level for the Earth, top of the model for simulations. It can be computed by integration of lateral density variations across the mantle model. Additional contributions arise from lateral mass heterogeneities produced by deflections of interfaces, surface and CMB in particular. These deflections are not explicit in the models (which assume spherical boundaries), but can be computed from element $\tau_{rr}$ of stress tensor $\boldsymbol{\tau}$ at interfaces, where $r$ is the radial coordinate. Geoid computation is an intrinsic capacity of both codes *CitcomS* and *StagYY*. It follows Zhang and Christensen (1993), and is implemented as in Zhong et al. (2008). Interface topographies and density heterogeneities are projected on spherical harmonics $Y_l^m$. Surface geoid and interface topographies are computed for each spherical harmonic degree $l$ and order $m$ using the flow solver, complemented by an effective pressure term that accounts for self-gravitation. Because of the large viscosity lateral variations, this procedure is required over simpler methods based on geoid kernels assuming radial viscosity distributions (Richards and Hager, 1984; Ricard et al., 1984; Hager et al., 1985). The reader is referred to Zhong et al. (2008) for a more detailed description of the method. Geoid spherical harmonic coefficients $c_{l,m}$ and $s_{l,m}$ are computed at each time step. Geoid undulations $N(\lambda, \phi)$ can then be expressed as a function of latitude $\lambda$ and longitude $\phi$ as:

$$N(\lambda, \phi) = R \sum_{l=2}^{\infty} \sum_{m=0}^{l} \left[ c_{l,m} \cos m\phi + s_{l,m} \sin m\phi \right] P_l^m(\sin \lambda), \tag{1}$$

where $R$ is Earth's radius, and $P_l^m$ the associated Legendre polynomial of degree $l$ and order $m$.

### 2.2.1 Geoid in model MF

Model MF is forced at the surface by the plate model. To compute the geoid at a given time step, we solve the Stokes flow with self-gravitation at this time-step with a free-slip condition at the surface as usually done in plate-driven models (Steinberger, 2016; Flament, 2019; Mao and Zhong, 2021). Two geoid outputs are computed for this model. The first one, called "Total geoid" is computed as described above, retaining complete density and viscosity heterogeneities of model MF without any modifications to the density or viscosity fields. The computation of the geoid is very sensitive to large lateral viscosity variations in the mantle (Čadek and Fleitout, 2003; Flament, 2019). Model MF is driven by a plate reconstruction model, updated every

1 Myr, which notably imposes the positions of viscous slabs. The update of the slab positions strongly affects this "Total geoid", creating discontinuities in the time evolution of the geoid. To tackle this issue, we compute a second geoid, called "No LVVs geoid", for which we discard density and viscosity lateral variations in the upper 350 km. This is done by setting all temperatures at a given depth in this range to their mean value before computing the geoid with the Stokes flow solver with self-gravitation. The density and viscosity distribution in the mantle below 350 km is not modified. In rare cases, the flow solver does not converge. Such a case is found at time -190 Myr in model MF for the "No LVVs geoid" case. We then simply interpolate the geoid computed at times -185 and -195 Myr. The "Total geoid" is rather different from the "No LVVs geoid" because cold slabs in the upper mantle strongly increase the local viscosity, which has a large effect on the surface dynamic topography it produces, hence on the geoid (Flament, 2019). Radial viscosity profiles are classically used to compute the geoid using geoid kernels (Richards and Hager, 1984; Rouby et al., 2010; Steinberger et al., 2019b). In our plate-like models, the largest lateral variations of viscosity occur in the upper mantle. Removing the effects of these lateral variations in the upper mantle thus enables us to compute a geoid that is closer to the one computed from radial geoid kernels. A third geoid has been computed by cancelling only the lateral variations of density above 350 km depth. The geoid produced in this case is very close to the "Total geoid" and the TPW path does not significantly differ. We thus discarded this case for this study.

### 2.2.2    Geoid in model MC

Model MC is fully self-consistent: no forcing is imposed at the surface, where a stress-free boundary condition is applied. The "Total geoid" computed in MC evolves smoothly. It is thus not necessary to remove the effect of lateral variations in the upper mantle to obtain a smoother geoid evolution as it was the case in model MF, and only the "Total geoid" is computed for this model. The geoid computation is done within a benchmarked module of the *StagYY* code previously used for the Earth's case (Cammarano et al., 2011; Guerri et al., 2016) as well as for Venus (Rolf et al., 2018). Though the time step between two successive snapshots of the CMB heat flux is 1 Myr in model MC, we only have access to the geoid every 5 Myr to 11 Myr. In order to obtain the position of the pole every 1 Myr, we performed a linear interpolation of the computed geoids.

### 2.3    True polar wander implementation

TPW is governed by the conservation of Earth's angular momentum, yielding Liouville's equation (Ricard et al., 1993). Here we use a simplified approach to compute TPW by considering that Earth's spin axis aligns instantaneously with the maximum inertia axis (Steinberger and O'Connell, 1997; Zhong et al., 2007). This method neglects the viscous delay due to the Earth's equatorial bulge adjustment (Cambiotti et al., 2011). Those principal axes are obtained from the geoid computed in the mantle convection simulations. The inertia tensor $\mathbf{I}$ due to mass redistribution in the mantle is built from the degree 2 coefficients of the geoid, following MacCullagh's formula (Schaber et al., 2009):

$$\mathbf{I} = MR^2\sqrt{\frac{5}{3}}\begin{pmatrix} \frac{c_{2,0}}{\sqrt{3}} - c_{2,2} & -s_{2,2} & -c_{2,1} \\ -s_{2,2} & \frac{c_{2,0}}{\sqrt{3}} + c_{2,2} & -s_{2,1} \\ -c_{2,1} & -s_{2,1} & -2\frac{c_{2,0}}{\sqrt{3}} \end{pmatrix}, \tag{2}$$

where $M$ and $R$ are Earth's mass and radius, respectively. The principal inertia axes are then obtained through a diagonalization of this matrix. The maximum inertia axis corresponds to the largest eigenvalue, while the two equatorial principal axes correspond to the smallest and intermediate eigenvalues. Computing the maximum inertia axis gives two new poles, one on each side of the planet. Which of the two poles is the "north pole" is arbitrary, and is chosen at the beginning of the simulation.

TPW is then implemented iteratively by rotating the mantle at each time step to ensure that the spin axis follows the position of the maximum inertia axis. The rotation direction is chosen so that the new north pole remains in the same hemisphere as the previous one, effectively limiting TPW amplitudes to a maximum of 90° per iteration. Since TPW is governed by the geoid, we have two different TPW paths for model MF. In the following, the TPW associated with the total geoid is called "Total TPW", while TPW associated with the no LVVs geoid is called "No LVVs TPW".

This TPW implementation corresponds to a change in the reference frame in which data are represented. This new reference frame is permanently wandering with respect to the initial simulation frame, we thus call it the wandering frame in the following. The simulation is not related to any forcing in model MC other than the initial conditions. The only preferential relation between the simulation frame and the wandering frame is thus through the initial conditions in model MC. In contrast, the simulation frame corresponds to the mantle reference frame of the plate reconstruction in model MF. If both the plate reconstruction and the resulting mantle convection simulation were perfectly tuned to the Earth, the wandering frame should merge with the simulation frame (to within a rotation in longitude) at the end of the simulation, *i.e.,* for the present time. In past times, the two frames are expected to diverge because of the effect of TPW. In practice, a non-negligible shift exists between the simulation and the wandering frames at the end of the simulation in both the "Total TPW" and "No LVVs TPW" cases.

An alternative case is considered for model MF by rotating the model results into the paleomagnetic reference frame. This case is more comparable to the work of Dannberg et al. (2024), in which the plate reconstruction of Merdith et al. (2021) is used in the paleomagnetic reference frame to drive convection. To do so, we compute the net rotations from the reconstruction of Merdith et al. (2021) in the paleomagnetic reference frame using the software *GPlates*, and rotate the model outputs accordingly. In this case, the wandering frame is the paleomagnetic reference frame. This alternative correction enables us to position the surface of the mantle model in a reference frame in which the magnetic dipole as seen by paleomagnetism is aligned with the z-axis. In this reference frame, the surface and interior undergoes solid body rotation. As a result, in our case MF*, the rotation of the deep mantle is the combination of the net rotation of the deep mantle in model MF and the solid body rotation to set the model in the paleomagnetic reference frame. This behaviour is what would be expected if all the net rotations of the surface in the paleomagnetic reference frame were due to a solid body rotation of the mantle. In practice, this net rotation of the surface is an undistinguishable combination of solid body rotation (part of which is due to TPW) and differential rotation between the lithosphere and the mantle. By forcing all the surface net-rotations to entrain a solid body rotation in case MF*, the coupling between the surface net rotations and the deep mantle net rotations is overestimated. We note that the coupling between the net rotations of the lithosphere and the deep mantle is also likely overestimated in Dannberg et al. (2024) due to the absence of continental keels and stress-dependent rheology in their models (J. Dannberg, personal communication, February 22, 2024). Contrary to the TPW corrections based on the geoid, this alternative approach is not a self-consistent way of rotating the model outputs in the reference frame of the spin axis. It moreover has the disadvantage of imposing lateral displacements

| Name | Surface conditions | Model duration | $\Delta t_{snap}$ | $N_{snap}$ | $\delta\rho_p$ | $\delta\rho_c$ | Correction |
|------|--------------------|----------------|-------------------|------------|----------------|----------------|------------|
| MF0 | Plate reconstruction | 1000 Myr | 5 Myr | 201 | +95.2 kg m$^{-3}$ | -140 kg m$^{-3}$ | None |
| MF1 | Plate reconstruction | 1000 Myr | 5 Myr | 201 | +95.2 kg m$^{-3}$ | -140 kg m$^{-3}$ | Total TPW |
| MF2 | Plate reconstruction | 1000 Myr | 5 Myr | 201 | +95.2 kg m$^{-3}$ | -140 kg m$^{-3}$ | No LVVs TPW |
| MF$^*$ | Plate reconstruction | 1000 Myr | 5 Myr | 201 | +95.2 kg m$^{-3}$ | -140 kg m$^{-3}$ | Paleomagnetic |
| MC0 | Free-slip | 831 Myr | 1 Myr | 832 | +137 kg m$^{-3}$ | -225 kg m$^{-3}$ | None |
| MC1 | Free-slip | 831 Myr | 1 Myr | 832 | +137 kg m$^{-3}$ | -225 kg m$^{-3}$ | Total TPW |

**Table 1.** Characteristics of the six cases analysed in this study. $\Delta t_{snap}$ is the time step between two successive snapshots of the CMB heat flux. Note that in model MC the time step between two geoid outputs is larger than $\Delta t_{snap}$. $N_{snap}$ is the total number of CMB heat flux snapshots. $\delta\rho_p$ an $\delta\rho_c$ are the excess density of chemical piles and deficit density of continents, respectively. The value of $\delta\rho_c$ in model MF is an average over all continents (see Flament et al. (2014) for more details on the modelling of continents in model MF). The durations of cases MC0 and MC1 correspond to the total duration of model MC minus the 300 first Myr (relaxation time).

of the deep mantle, entrained by a net rotation of the surface (Rudolph and Zhong, 2014; Müller et al., 2022). We nevertheless consider this case as we are mostly interested in the latitudinal distribution of the CMB heat flux, and this correction gives the
model outputs in a reference frame in which continents have the correct latitudes regarding paleomagnetic constraints.

In total, six cases are considered in the subsequent analysis of CMB heat flux patterns. Their characteristics are summarized in Table 1. Cases MF0 and MC0 are directly derived from models MF and MC, ignoring TPW. Cases MF1 and MC1 are obtained by correcting for the "Total TPW" computed from the "Total geoid" in both models. Case MF2 is derived from model MF corrected from the "No LVVs TPW", computed from the "No LVVs geoid". Case MF$^*$ is obtained from model MF by
rotating the outputs in the original paleomagnetic reference frame of the plate reconstruction.

Spherical harmonic transforms and rotations are performed using the library SHTns (Schaeffer, 2013). SHTns provides an efficient implementation of spherical harmonic rotations based on the stable recursive evaluation of Wigner's d-matrix proposed by Gumerov and Duraiswami (2015), which is accurate up to very large degrees ($> 10^4$).

## 2.4    Principal component analysis of CMB heat flux

We use a principal component analysis (PCA) to obtain the dominant heat flux patterns at the bottom of the mantle in the different models. PCA is a data analysis tool that can be applied to a dataset comprising several observations, each observation depending on several variables. It is used to express the dataset in a new orthonormal basis in order to limit the number of variables needed to explain the data. This is done by computing new variables (called principal components), which are combinations of the initial variables. A full mathematical description of PCA theory is given by Abdi and Williams (2010);
see also Pais et al. (2015) for an application to core flows and details of the method. Considering a data set containing $I$ observations described by $J$ variables, the PCA consists in a singular value decomposition of the $I \times J$ data matrix $\mathbf{D}$ as

$$\mathbf{D} = \mathbf{WSP}, \tag{3}$$

where $\mathbf{W}$, $\mathbf{S}$ and $\mathbf{P}$ have respective dimensions $I \times K$, $K \times K$ and $K \times J$, with $K = \min(I, J)$ the rank of the data matrix. $\mathbf{P}$ is a basis of $K$ new variables (or principal components), called $p_k$ with $k \in [\![1 \; ; \; K]\!]$, which are linear combinations of the initial $J$ variables. The amount of data variance explained by $p_k$ decreases with increasing $k$. This variance is quantified by a score, called $s_k$, corresponding to the singular values in the diagonal matrix $\mathbf{S}$. The square of $s_k$ gives the variance explained by $p_k$. The projections of the $I$ observations on this new basis are stored in the $\mathbf{W}$ matrix.

In the framework of this study, PCA is used to obtain the principal components corresponding to heat flux patterns that explain most of the heat flux signal at the CMB as a function of time. The data set consists in the spherical harmonics coefficients of $q_{CMB}$ for each snapshot of the mantle convection model, truncated at a maximum degree $l_{max} = 50$. PCA requires data to be centred, which means in our case that the mean of each spherical harmonic coefficient on the whole time series has to be removed. This operation is equivalent to removing the mean heat flux pattern $\overline{q}(\lambda, \phi)$ from the time series. Following previously described notations, the $k^{th}$ PCA component consists in a $J$-dimension vector $p_k(l, m)$ of spherical harmonic coefficients, a score called $s_k$, and a time-dependent weight $w_k(t)$. We note $\tilde{p}_k(\lambda, \phi)$ the heat flux pattern reconstructed from the $p_k(l, m)$ spherical harmonic coefficients. For each component, a time-dependent amplitude $A_k$ can be defined as $A_k(t) = w_k(t) \times s_k$. Because of data centring, the heat flux patterns of the PCA components have to be interpreted as perturbations to the mean heat flux pattern.

Once PCA is performed, time-dependent CMB heat flux maps can be reconstructed as:

$$q_{CMB}(\lambda, \phi, t) = \overline{q}(\lambda, \phi) + \sum_{k=1}^{K} w_k(t) \, s_k \, \tilde{p}_k(\lambda, \phi). \tag{4}$$

The first components give the highest contribution to the full variability of CMB heat flux. They provide plausible CMB heat fluxes that can also be applied as boundary conditions in dynamo calculations.

## 3 Results

### 3.1 Model descriptions

Let us first present and discuss important characteristics of the two mantle simulations we exploit. The radial profiles of viscosity and temperature in model MF and MC are shown in Fig. 1. The steeper temperature gradient at the base of the mantle in model MF imposes a CMB heat flux about 4 times larger in model MF than in model MC. Selected snapshots of topography, basal heat flux, and geoid undulations, are shown in Fig. 2 and Fig. 3 for models MF and MC, respectively.

Both models reproduce the well-known bimodal topography of the Earth, reflecting the difference between continental and oceanic lithosphere. Oceanic ridges and trenches are well marked. Some plate boundaries can be easily recognized in line "0 Myr" of model MF in Fig. 2, and topography snapshots at -300 Myr, -600 Myr, and -900 Myr are identical (to within a 180° rotation in longitude) to the "No Net Rotation" plate reconstruction shown in Fig. 3 of Müller et al. (2022). Let us recall that mantle simulation MF is driven by plates while plates spontaneously form and move in mantle simulation MC, and note that MC plates look similar to those of model MF (Fig. 3).

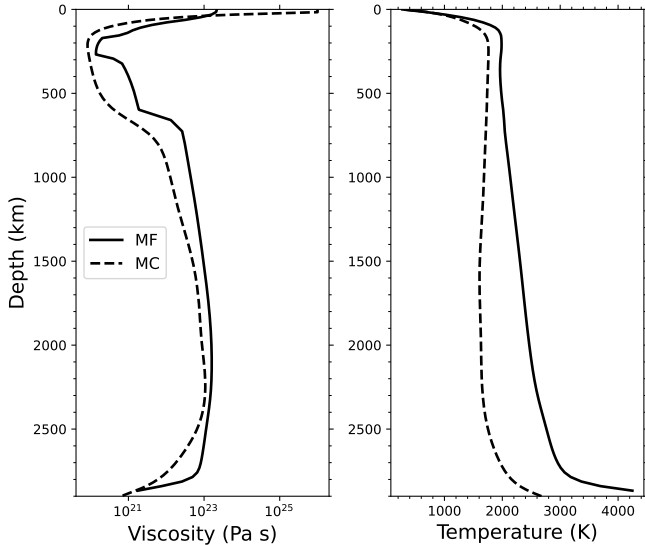

**Figure 1.** Viscosity and temperature profiles in model MF and MC. The profiles in model MF are the present-day (t = 0 Myr) horizontal-averaged profiles. The profiles in model MC are time-averaged and horizontal-averaged profiles. Viscosity profiles are computed from pressure and temperature profiles. Chemical anomalies are not taken into account here (continents and chemical piles). Model MC uses the Boussinesq approximation, while model MF uses the extended-Boussinesq formulation in which the effect of compression is considered, hence the larger lower mantle temperature in model MF.

CMB heat flux maps of both models show large-scale variations that strongly correlate with the presence of basal chemical piles (delineated by black lines). The heat flux is low beneath these piles, which act as thermal insulators. In both models, chemical piles are shaped and pushed by cold slabs that reach the CMB. The CMB heat flux is dominated by large scale heterogeneities as shown by the spectra in Fig. 4. The CMB heat flux in model MC is mostly dominated by the degree 2, while it is dominated by degrees 2, 3 and 6 in model MF.

Movies showing the time evolutions of the fields shown in Fig. 2 for model MF and Fig. 3 for model MC are available as supplementary materials, as well as movies of the CMB heat flux in the different cases. The full snapshots of the geoids, the CMB heat flux, the topography and the composition at the CMB are also available in a hdf5 format (every 2 Myr for MC0 and MC1); as well as scripts to perform the PCA and the TPW correction from the snapshots (*https://doi.org/10.5281/zenodo.10523321*).

### 3.2 Geoid

Our study requires computing the geoid in the models, in order to deduce the resulting TPW. The geoid stems from a delicate balance between bulk density heterogeneities and flow-induced interface undulations. The geoid computed in model MC (first column in Fig. 3) and the "Total geoid" in model MF (first column in Fig. 2) have similar amplitudes. They both display small-scale negative anomalies above active subduction zones. In model MF, subductions are often associated with broader negative

geoid anomalies. In model MC, subducting slabs are on the contrary associated with positive anomalies on larger scales (see snapshots at -505 Myr and - 757 Myr in Fig. 3). It is notably the case for the present day time on Earth, with geoid highs above

320 Andine and Indonesian subduction zones (Crough and Jurdy, 1980; Hager, 1984). The chemical piles are mostly associated in both cases with large-scale negative geoid anomalies (see snapshots at -600 Myr in Fig. 2 and snapshots at -254 Myr in Fig. 3). At the beginning of model MF, the piles are associated with positive anomalies in the "Total geoid", as can be seen in the snapshot at -900 Myr. This positive signal above the piles only lasts for the first 150 Myr of the simulation. This change of sign in the geoid anomalies above the piles could thus be an effect of the initial conditions. The correlation between the piles

and the geoid is stronger in model MC, in which the density excess of the piles is larger than in model MF. In both models, the piles are shaped by the subducting slabs as they reach the lower mantle. As on Earth, deciphering the role of deep hot domes and subducting plates in the geoid signal thus remains a fundamental issue (Rouby et al., 2010). This is beyond the scope of this study.

The surface conditions of model MF are updated every 1 Myr. The "Total geoid" being strongly affected by the positions of

330 subducting slabs, the update of the surface conditions implies fast variations of the geoid from one snapshot to the next. We evaluate this effect by computing the "No LVVs geoid", shown in the second column of Fig. 2. We recall that this alternative geoid is computed after removing density and viscosity lateral variations in the upper 350 km of the mantle. As expected, the sharp signature of subductions disappears. This "No LVVs geoid" has a much smoother time evolution, since it is not affected by the high-frequency surface updates. The large-scale pattern is also modified, and the geoid amplitudes are larger than in

the "Total geoid" case. Removing the contribution from the shallower mantle also increases the correlation between the geoid and the chemical piles, with mostly positive geoid anomalies appearing above the piles. This correlation between the piles and positive geoid anomalies is opposite to what is observed in model MC and in model MF with the "Total geoid". It is however more representative of the observations on the present-day Earth, with two broad positive geoid anomalies above the LLVPs. Despite these differences in behaviour, the "No LVVs geoid" and the "Total geoid" in model MF are generally similar at the

beginning of the simulation (see snapshot at -900 Myr) and to a lesser extent at the end of the simulation (see snapshot at 0 Myr). The impacts on the CMB heat flux of the different geoid behaviours are discussed in Sect. 4.

### 3.3 TPW correction

TPW is applied by computing the successive positions of the maximum inertia axis and rotating the simulation frame to align this axis with the spin axis. The successive positions of the maximum inertia axis in the simulation frames of models MF and

345 MC for cases MF1, MF2 and MC1 are shown in Fig. 5. These successive positions represent the TPW path in each case. The wandering frames would be identical to the simulation frames if the maximum inertia axis stayed fixed at either pole in the simulation frames. The rotation between the wandering frame and the simulation frame is the largest when the spin axis plots at the equator.

Let us analyse the TPW paths for model MF, computed either from the "Total geoid" (MF1) or the "No LVVs geoid" (MF2).

The present-day position (black-circled magenta disk in Fig. 5) of the maximum inertia axis is similar in both cases. Since model MF satisfies plate configuration at $t = 0$, this axis should plot at one pole of the simulation if it faithfully reproduced

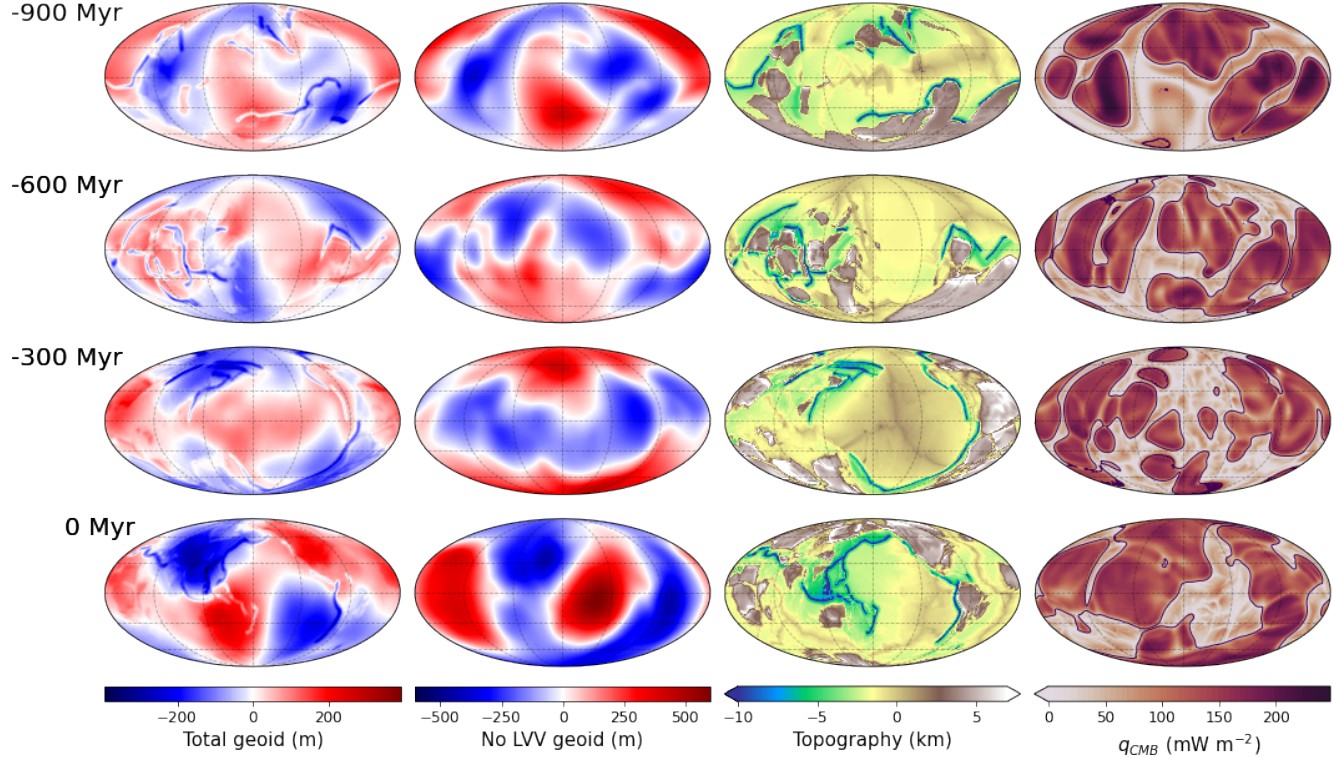

**Figure 2.** Model MF. Maps of geoid undulations (first column: "Total geoid"; second column: "No LVVs geoid"), surface topography (third column), and CMB heat flux (last column) at selected times. The maps are shown in a Mollweide projection. Black lines in the heat flux maps delineate the edges of basal chemical piles. The heat flux is low beneath chemical piles.

actual mantle configuration. This is not precisely the case, but we observe that the spin axis for MF1 mostly remains at high latitudes after -600 Myr, implying a relatively small correction between the wandering frame and the simulation frame. Though the final position of the maximum inertia axis in MF2 is similar to that in MF1, the paths followed by the poles quickly diverge
back in time. Contrary to MF1, the maximum inertia axis stays at relatively low latitudes for the whole simulation in MF2. During the first 150 Myr of the simulation, the poles of the wandering frames in MF1 and MF2 are almost antipodal, meaning that the inertia axes are almost aligned. This is consistent with the similar geoid patterns in both cases at -900 Myr shown on Fig. 2. Note that the TPW path for MF1 are very irregular due to a large time variability in the "Total geoid", induced by the updates of subduction zones every 1 Myr. This scatter of the position of the spin axes mostly implies erratic deviation from a mean
path. It can however trigger larger discontinuities. Such a large deviation occurs at -260 Myr. A second occurrence of spin-axis instability occurs between -425 Myr and -405 Myr. Interestingly, this unstable period starts with an inertial interchange TPW (IITPW) event between -425 Myr and -420 Myr. This event is highlighted by a black arrow in Fig. 5. An IITPW event occurs when the maximum inertia axis and the intermediate inertia axis switch order, resulting in a $\sim 90°$ rotation between the two successive time steps. Such kind of events have been suggested as an explanation for fast apparent polar wander in the

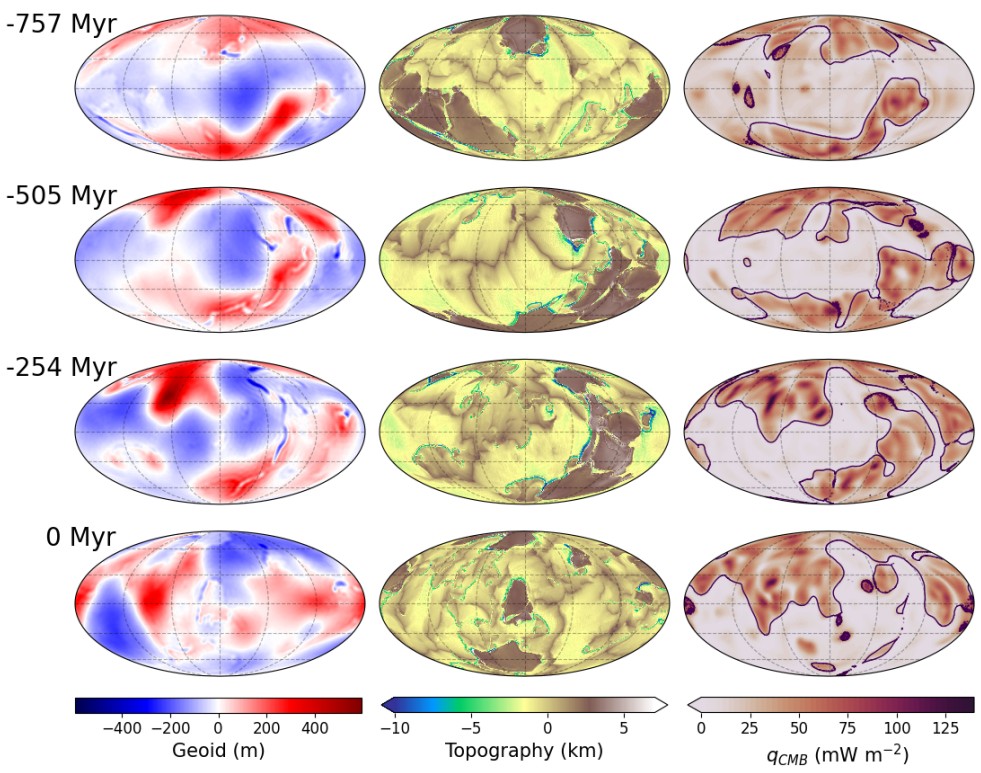

**Figure 3.** Model MC. Maps of geoid undulations (left), surface topography (centre), and CMB heat flux (right) at selected times. The maps are shown in a Mollweide projection. Black lines in the heat flux maps delineate the edges of basal chemical piles. Heat flux is low beneath chemical piles.

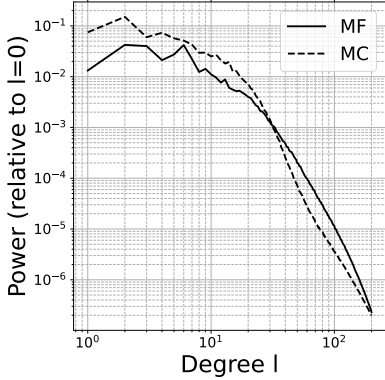

**Figure 4.** Time-averaged spherical harmonic spectra of the CMB heat flux in model MF and MC. The spectra are given relative to the power of the l=0 coefficient.

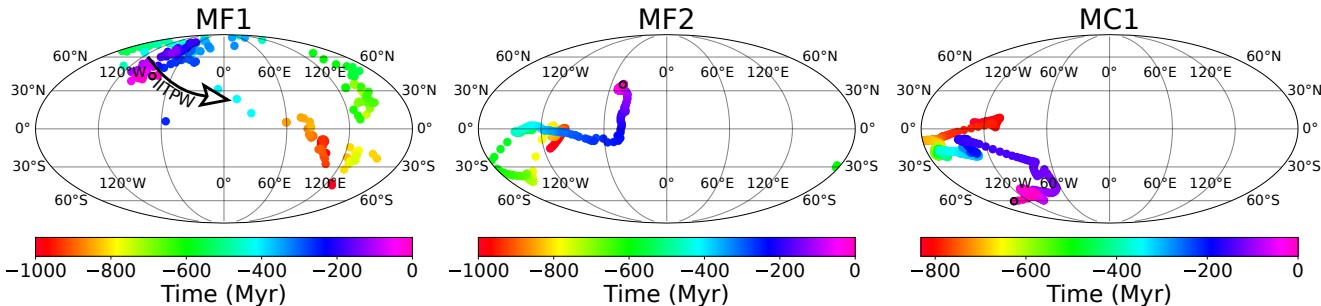

**Figure 5.** TPW paths in cases MF1, MF2 and MC1 in a Mollweide projection. Colour disks represent the successive positions of the maximum inertia axis in the simulation frames. The colour scale gives the time before the end of the simulation in Myr. Black-circled magenta disk shows the position of the maximum inertia axis at the end of the simulation. The Inertial Interchange TPW event (IITPW) in case MF1 is highlighted by the black arrow.

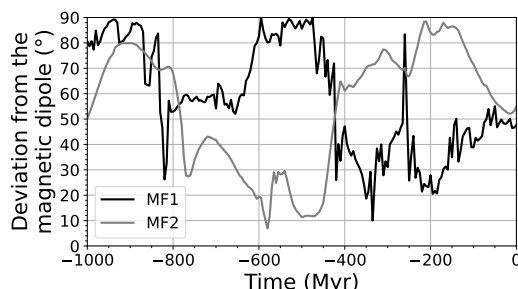

**Figure 6.** Angular distance between the maximum inertia axis computed in MF1 and MF2, and the axis of the magnetic dipole (corresponding to the z-axis of the paleomagnetic reference frame of Merdith et al. (2021)). A small angle means a high consistency between the geoid and the position of the magnetic pole.

Cambrian and Ediacaran periods (Kirschvink et al., 1997; Robert et al., 2017). No significant TPW event is observed in MF2 at this time, suggesting that this IITPW event in MF1 is triggered by shallow heterogeneities.

  Turning to model MC, we recall that this model is not related to any plate reconstruction. Hence no particular relationship is expected between the simulation frame and the wandering frame. However, the two frames are linked due to the relationship between chemical piles and the geoid in the model. Chemical piles are introduced at the equator at the beginning of the

370 simulation, and remain at low latitudes by spreading around the equator. These piles are mostly associated with geoid lows throughout the simulation. They thus tend to move from low latitudes towards high latitudes due to the TPW correction. As a result, the maximum inertia axis tends to form a $\sim 90°$ angle with the $z$-axis of the simulation frame as observed in Fig. 5.

  In cases MF1 and MF2, the positions of the maximum inertia axes can be compared to the position of the pole in the paleomagnetic reference frame. By doing so, we can evaluate the consistency between the paleomagnetic constraints and the

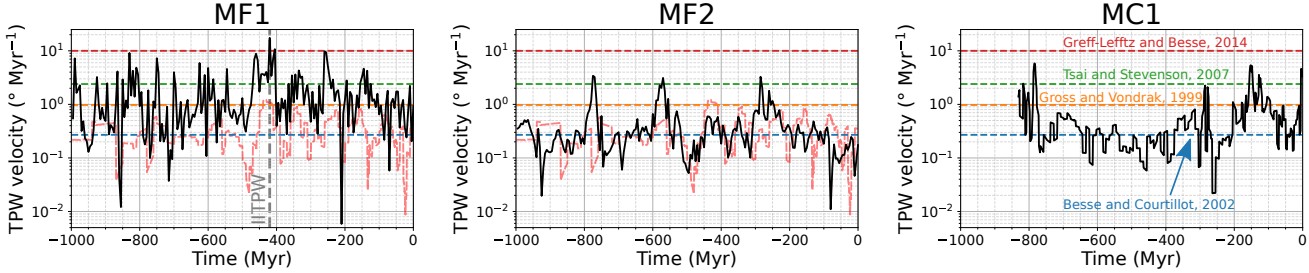

**Figure 7.** TPW velocity in $^\circ$ Myr$^{-1}$ as a function of time in cases MF1, MF2, and MC1. The light red dashed line in cases MF1 and MF2 show the displacement rate of the spin axis in case MF$^*$. This displacement rate corresponds to the velocity at which the magnetic dipole axis rotates in the reference frame of model MF. The horizontal dashed lines give some values from the literature: velocities during past TPW events in the last 200 Myr (Besse and Courtillot, 2002); present-day TPW velocities (Gross and Vondrák, 1999); maximum TPW velocity value for the present-day mantle (Tsai and Stevenson, 2007); maximum modelled TPW velocity for an inertia interchange event (Greff-Lefftz and Besse, 2014).

geoid produced by the mantle convection model rotated in the paleomagnetic reference frame as it is done for case MF$^*$. We recall that running such a mantle convection model is not desirable as it implies large lateral movements of the whole mantle (Müller et al., 2022). It however has the advantage of ensuring the continent positions are consistent with the paleomagnetic constraints. The deviation between the positions of the inertia axis computed in MF1 and MF2 and the spin axis in case MF$^*$ can be used to evaluate the consistency between the moment of inertia and the paleomagnetic reference frame. If this deviation is small, the inertia axis and the magnetic dipole are nearly aligned as expected for the Earth. If the deviation is large, the inertia axis is not aligned with the magnetic dipole and the geoid is thus inconsistent with the orientation of the plate reconstruction. The angular deviations between the maximum inertia axes (computed in MF1 and MF2) and the spin axis in MF$^*$ are shown in Fig. 6. The deviation between the maximum inertia axis and the spin axis widely varies during the course of the simulation. The deviation is large for both MF1 and MF2 at the beginning of the simulation. Between -785 Myr and -425 Myr, the deviation is significantly smaller for MF2 than for MF1. For the last part of the simulation, MF1 shows a smaller deviation than MF2 except during the pole instability at -260 Myr. None of the "Total geoid" in case MF1 and the "No LVVs geoid" in MF2 thus give a geoid consistent with the position of the spin axis in case MF$^*$.

The time evolution of TPW velocities (in $^\circ$ Myr$^{-1}$) for the different cases are shown in Fig. 7. Those velocities are compared to values of TPW velocities from the literature represented as horizontal dashed lines: Besse and Courtillot (2002) gave a typical velocity of 0.27$^\circ$ Myr$^{-1}$ during TPW events in the last 200 Myr; Gross and Vondrák (1999) measured a present-day TPW velocity of 0.98$^\circ$ Myr$^{-1}$; Tsai and Stevenson (2007) found 2.4$^\circ$ Myr$^{-1}$ as a maximum TPW velocity reachable for the present-day mantle; Greff-Lefftz and Besse (2014) obtained TPW velocities up to 10$^\circ$ Myr$^{-1}$ during an IITPW event in simplified mantle convection models. The displacement rate of the spin axis of the wandering frame in case MF$^*$ is shown in case MF1 and MF2 as the light red curve. As for the TPW velocity, this rotation rate gives the rate at which the mantle material is rotated in latitude in case MF$^*$. This displacement rate is not affected by net rotations in longitudes occurring in case MF$^*$, as

these azimuthal rotations do not imply a displacement of the spin axis. In all our cases, TPW velocities are roughly contained between $0.01°$ $\text{Myr}^{-1}$ and $10°$ $\text{Myr}^{-1}$. Note that because of the 5 Myr time step between two successive snapshots in model MF, TPW velocities cannot be higher than $18°$ $\text{Myr}^{-1}$ for MF1 and MF2. Similarly in MC1, TPW velocities are limited to between $9°$ $\text{Myr}^{-1}$ and $18°$ $\text{Myr}^{-1}$ depending on the time step between two successive geoid snapshots. The average TPW

velocity is $\sim 1.79°$ $\text{Myr}^{-1}$ in MF1. This is much faster than in MF2 and MC1, with averaged TPW velocities $\sim 0.42°$ $\text{Myr}^{-1}$ and $\sim 0.58°$ $\text{Myr}^{-1}$, respectively. This faster TPW is related to the greater scatter of successive inertia axis positions in MF1 compared to MF2 and MC1. The faster TPW velocities are thus directly related to the updating of surface conditions that implies a large time variability of the "Total geoid" in the MF simulation. The highest peak at -420 Myr in MF1 reaches $17.5°$ $\text{Myr}^{-1}$. This corresponds to a rotation of $87°$ over 5 Myr (one time increment). Such a large rotation is due to the IITPW event

occurring between -425 Myr and -420 Myr. This velocity is greater than the one obtained by Greff-Lefftz and Besse (2014) for an IITPW event in simplified mantle models, although Robert et al. (2017) obtained similar values for the maximum velocity reached for a specific IITPW event during Ediacarian times. One should however keep in mind that we neglected the time delay due to the equatorial bulge adjustment in our TPW correction (Ricard et al., 1993).

The average rotation rate in MF* is $0.33°$ $\text{Myr}^{-1}$. Except for the velocity peaks around -770 Myr, -570 Myr, and -280 Myr,

the rotation rates in case MF* is similar in amplitude to the TPW velocity in MF2. The TPW velocity in MF1 is however consistently higher than the rotation rate of the spin axis in case MF*. The fastest displacement of the spin axis occurs between -440 Myr and -410 Myr. During this period, the spin axis moves at a rate of about $1°$ $\text{Myr}^{-1}$, corresponding to the present-day TPW velocity (Gross and Vondrák, 1999). This period of fast rotation in MF* coincides with a period of fast TPW in MF1, during which the IITPW event occurs. It also correlates with a relatively fast TPW period in MF2, though not as fast as other

TPW events. Though we compare the rotation rate of the spin axis in MF* to TPW velocities, we stress that the net-rotations in MF* are not directly TPW velocities. They strictly correspond to the displacement velocity of the magnetic dipole axis as defined in the reconstruction of Merdith et al. (2021) in the no-net-rotation reference frame of model MF.

### 3.4 CMB heat flux

#### 3.4.1 Time variability of large-scale patterns

Most studies exploring the effect of CMB heat flux heterogeneity on the geodynamo focused on large-scale patterns of spherical harmonic degrees 1 and 2 (Glatzmaier et al., 1999; Olson and Christensen, 2002; Kutzner and Christensen, 2004; Olson et al., 2010). These low degrees are strong in the averaged power spectrum of the CMB heat flux in both MF and MC models, notably the degree 2. We thus first examine the time evolution of these patterns in the different cases. The CMB heat flux is decomposed in spherical harmonics following

$$q_{CMB}(\lambda, \phi) = \sum_{l=0}^{\infty} \sum_{m=-l}^{l} z_{l,m} P_l^m(\sin \lambda) e^{im\phi}. \tag{5}$$

For $m = 0$, the imaginary part of the complex coefficients $z_{l,m}$ is null and we only consider the real part of the coefficient. For $m > 0$, the $z_{l,m}$ coefficients have both a real and imaginary parts. We thus compute the module of the coefficient, multiplied

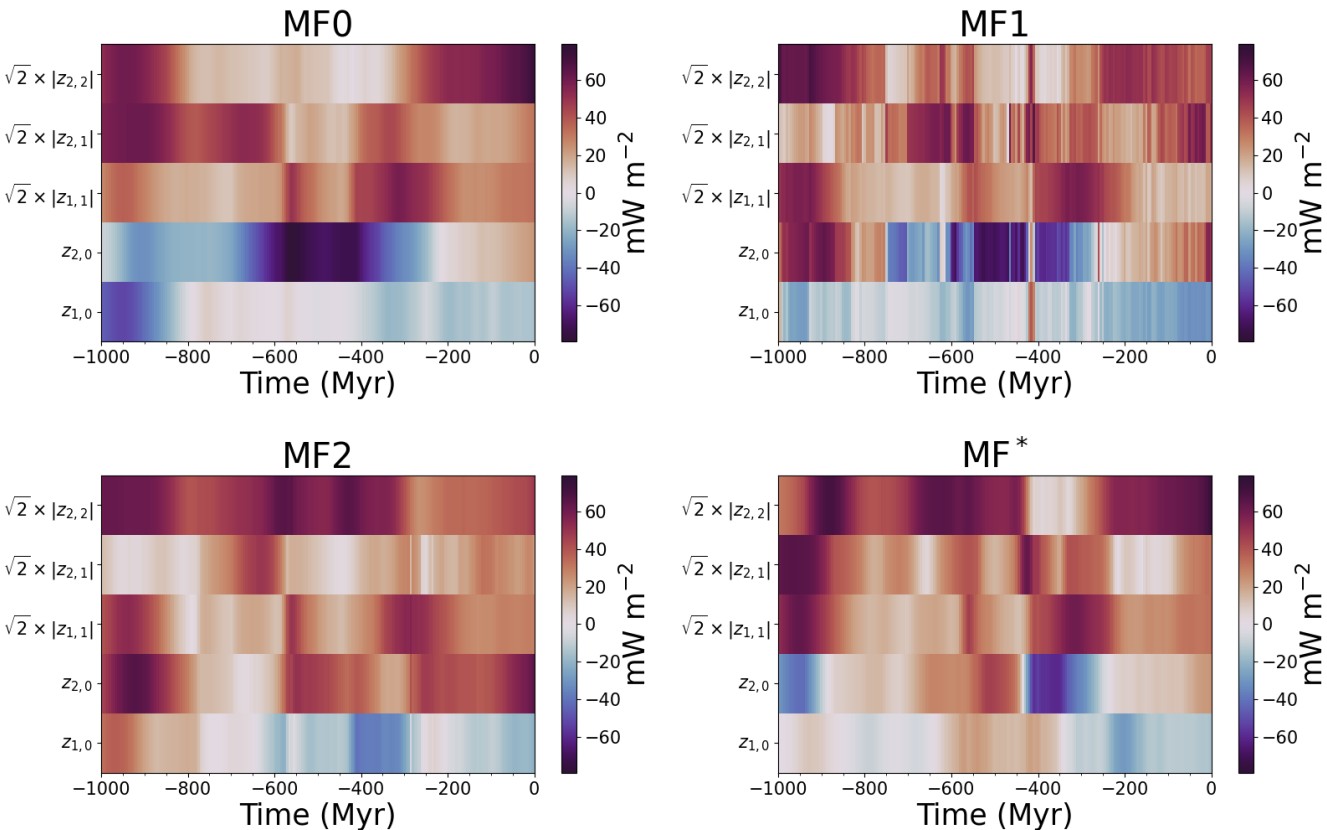

**Figure 8.** Time evolution of degree 1 and 2 spherical harmonic coefficients of the CMB heat flux in cases MF0, MF1, MF2 and MF*. Cases MF1 and MF2 are derived from MF0 by correcting for the TPW using the "Total geoid" and the "No LVVs geoid" in model MF respectively. Case MF* is derived from model MF0 by rotating model MF in the paleomagnetic reference frame of Merdith et al. (2021).

by a $\sqrt{2}$ factor to account for the $m < 0$ complex conjugate coefficients. The amplitude of degree 1 and 2 spherical harmonic coefficients of CMB heat flux are shown in Fig. 8 for cases MF0, MF1, MF2 and MF*, and in Fig. 9 for cases MC0 and MC1. As expected, low-degree patterns evolve rather smoothly, on mantle convection timescales of several hundreds of million years, in cases MF0 and MC0.

We have seen that CMB heat flux variations are strongly controlled by the distribution of basal chemical piles in our models. When piles are at high latitudes, for example at -600 Myr in case MF0 (see Fig. 2), the CMB heat flux is greater around the equator, yielding a negative $z_{2,0}$ coefficient. Inversely, with mostly equatorial piles at time -500 Myr, case MC0 has a strong positive $z_{2,0}$ coefficient at that time. The end of MF0 is dominated by the degree 2 and order 2 component. This pattern is expected at the end of the MF0 case, as model MF reproduces the present-day positions of the observed antipodal LLVPs below Africa and the Pacific (see Fig. 2).

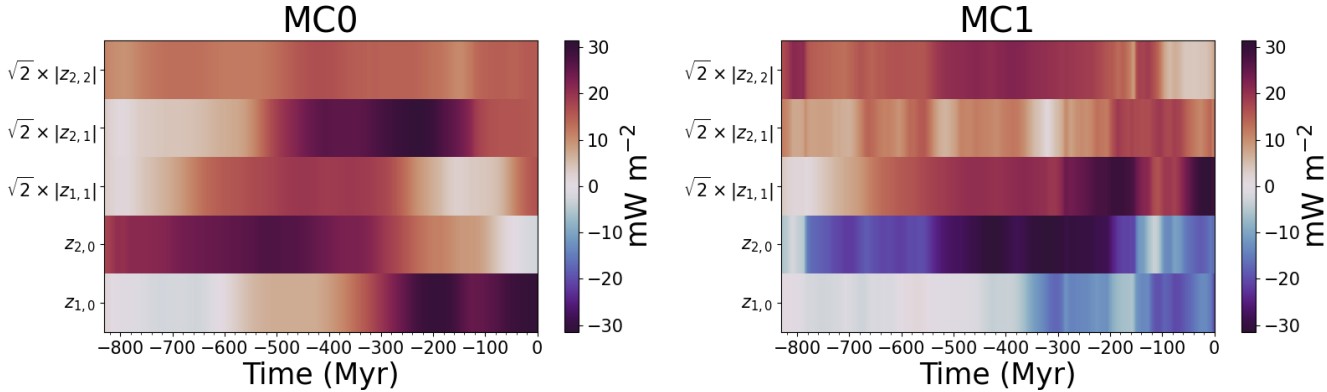

**Figure 9.** Time evolution of degree 1 and 2 spherical harmonic coefficients of CMB heat flux in cases MC0 and MC1. Case MC1 is derived from case MC0 by correcting it for the TPW.

True polar wander strongly impacts the behaviour of large-scale CMB heat flux patterns. We first note that TPW modifies the hierarchy of these patterns. For example, while the $z_{2,1}$ coefficient dominates between -425 Myr and - 210 Myr in MC0, its contribution becomes very weak in MC1. TPW also changes the sign of these coefficients. Spreading of the piles at the equator in model MC translates into a large positive $z_{2,0}$ coefficient (weaker heat flux around the equator) in MC0. This coefficient becomes negative in MC1 because the piles are moved towards higher latitudes by the TPW. Similarly, $z_{2,0}$ is negative during most of the time in MF0, but remains positive due to TPW in MF2. This is not the case in MF1, suggesting that the piles are more correlated with the "No LVVs geoid" than with the "Total geoid". The coefficients in MF* are the same as the one in MF0 at the end of the model, as there is no change of frame for the present-day in MF*. Changes in the hierarchy and sign of coefficients however occur for past times, notably before the large displacement of the spin axis between -450 Myr and -400 Myr. The degree 2 order 0 notably changes sign at -435 Myr, in the middle of this event.

More importantly, our study reveals that TPW changes large-scale patterns of CMB heat flux on timescales much shorter than typical mantle convection timescales. This is well illustrated by several sign reversals of the $z_{2,0}$ coefficient in case MF1, which occur over time lapses shorter than 10 Myr. This is the case during the IITPW event between -425 Myr and -420 Myr. The large number of fast changes in the spherical harmonic coefficients for MF1 is due to the rather rapid variations of the "Total geoid" computed from model MF. Nevertheless, slightly less rapid events are also visible in MF2, MF* and MC1. The variations of the degree 2 coefficients in MF* are particularly rapid around -435 Myr, despite this case having the lowest rotation rates of the spin axis. While the coefficients in MF* are very similar to case MF0 after this event, they largely differ before. This is the case of the degree 2 order 0 that changes sign, and of the degree 2 order 2 that suddenly weakens after this event.

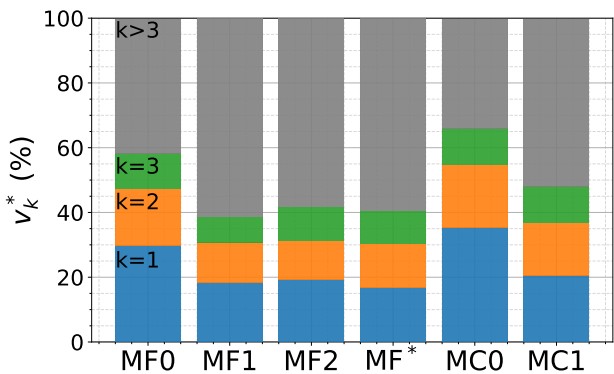

**Figure 10.** Variance explained by the first three PCA components normalized by the total variance in the different cases.

### 3.4.2 Principal component analysis of the CMB heat flux

As mentioned earlier, exploration of the effect of CMB heat flux heterogeneities on the geodynamo has previously mostly focused on degrees 1 and 2. An alternative is to explore the effect of heterogeneities inferred from seismic tomography of the lowermost mantle (Olson et al., 2010; Mound and Davies, 2023). However, the latter approach is only appropriate to describe the present core-mantle coupling, assuming the conversion of seismic tomography to temperature is well understood. In order to assess past geodynamo behaviour, one would like to test the effect of plausible past CMB heat flux patterns and amplitudes. Snapshots of the heat flux maps we computed could be used for this purpose. Another possibility is to examine what are the dominant heat flux patterns, and how they evolve in time. This is what we propose, using a principal component analysis (PCA).

PCA results consist in a set of components, ranked by decreasing contribution to the total CMB heat flux variations. Each component is described by a heat flux pattern and an amplitude. The amount of variance $v_k$ explained by the $k^{th}$ PCA component is given by the square of the associated score: $v_k = s_k^2$.

The variance explained by the first three PCA components is shown in Fig. 10. These components account for 58% and 66% of the total variance in cases MF0 and MC0, respectively. The amount of variance explained by the first three component is lower in the cases rotated in the spin-axis frame. For model MF, the first three components explain 39%, 42% and 40% of the total variance in cases MF1, MF2 and MF* respectively. This value reaches 48% for MC1. These lower values in rotated cases are due to the addition of a source of time variability in the CMB heat flux due to the net rotations occurring in the wandering frame. Among the rotated cases, the explained variance is the lowest in MF1. This can be related to rotation rates of the wandering frame, which are the fastest on average in case MF1, increasing the time variability.

The patterns of the first three PCA components are shown in Fig. 11 for cases MF0, MF1, MF2 and MF*, and in Fig. 12 for cases MC0 and MC1. Those patterns are perturbations to the average heat flux pattern, also shown in the figures for each case.

The associated amplitudes of the first three PCA components are shown in Fig. 13 for cases MF0, MF1, MF2 and MF*, and in Fig. 14 for cases MC0 and MC1. The amplitudes of the first PCA components mostly vary on large time scales. However,

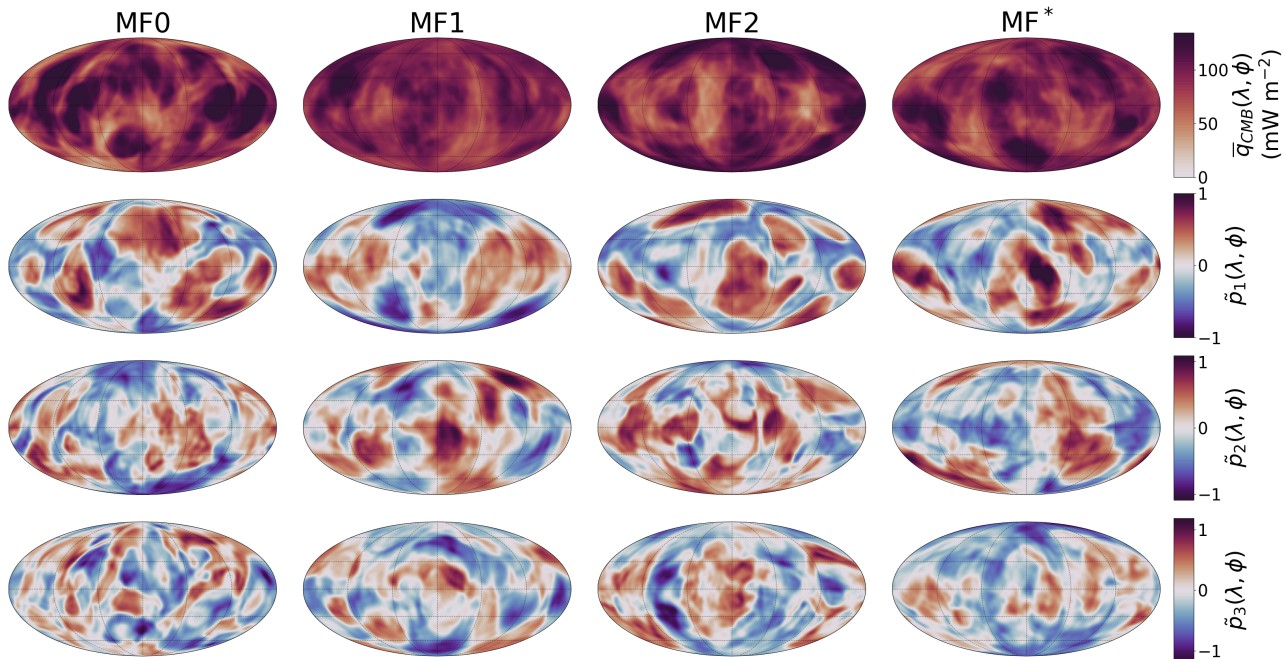

**Figure 11.** Patterns of the first three PCA components of the CMB heat flux $\tilde{p}_1$, $\tilde{p}_2$ and $\tilde{p}_3$ in cases MF0, MF1, MF2 and MF*. The averaged heat flux pattern is given in the top line for each case. The maps are shown in a Mollweide projection.

the rotated cases display amplitude variations at much higher frequencies, due to the time variability added by TPW. This is particularly visible for case MF1, which displays the largest rotation rate of the wandering frame.

## 4 Discussion

Recent models offer the possibility to study mantle convection related to plate tectonics on timescales of the order of 1 Gyr. These models reproduce plate tectonics either self-consistently (Coltice et al., 2019) or using plate reconstructions that have
485 been extended to cover the last 1 Gyr by recent works (Merdith et al., 2021). Both kinds of models enable to study the relation between surface tectonics and the structure of the lower mantle (Cao et al., 2021; Flament et al., 2022). The CMB heat flux is of particular interest for the geodynamo (Glatzmaier et al., 1999; Kutzner and Christensen, 2004; Olson et al., 2010). However, these models have to be rotated in a reference frame that preserves the spin axis in order to study the relation between CMB heat flux and the geodynamo. In this study, we analysed two such large recent mantle convection simulations by correcting
these models for the TPW in order to obtain the CMB heat flux in the frame appropriate for the geodynamo.

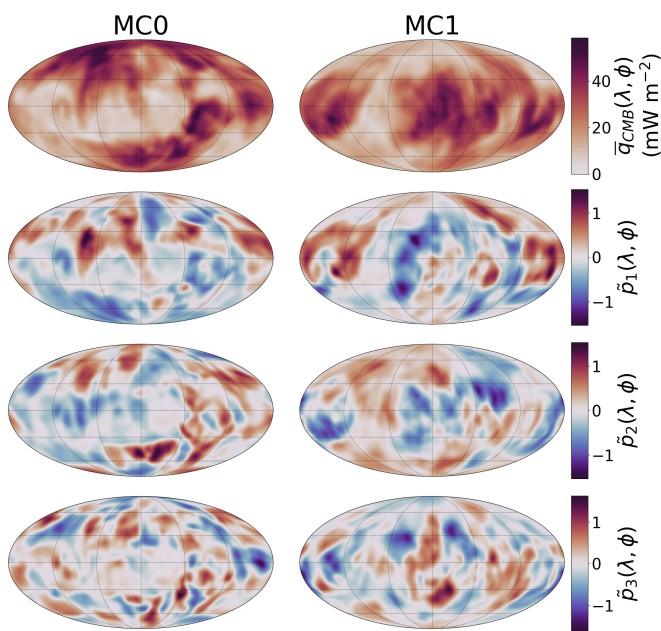

**Figure 12.** Patterns of the first three PCA components of the CMB heat flux $\tilde{p}_1$, $\tilde{p}_2$ and $\tilde{p}_3$ in cases MC0 and MC1. The averaged heat flux pattern is given in the top line for each case. The maps are shown in a Mollweide projection.

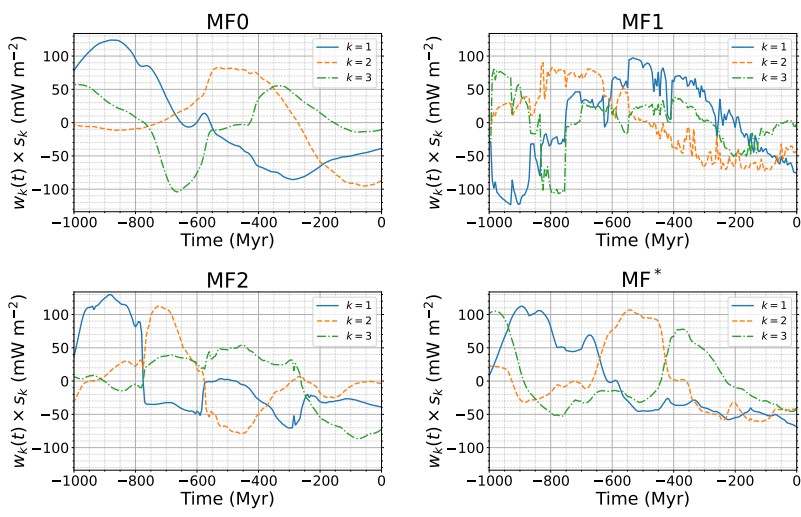

**Figure 13.** Time-evolution of the amplitudes of the first three PCA components of the CMB heat flux for cases MF0, MF1, MF2 and MF$^*$.

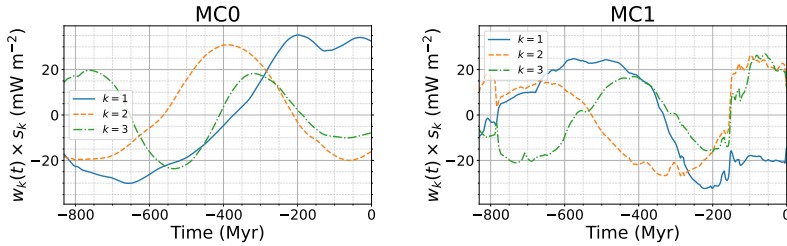

**Figure 14.** Time-evolution of the amplitudes of the first three PCA components of the CMB heat flux for cases MC0 and MC1.

## 4.1 Impact of TPW on the CMB heat flux

In the two mantle convection models studied here, we find that correcting for the TPW (or rotating the model in a paleomagnetic reference frame in case MF*) induces fast variations of the CMB heat flux, even at low spherical harmonic degrees. These fast variations, illustrated in Fig. 8 and Fig. 9, can induce changes in the hierarchy of spherical harmonic modes, and in the sign of the coefficients. The TPW induced variations do not originate from a change in the flux pattern but rather from a global rotation of a given pattern. Though mantle convection is completely unaffected by global rotations of the mantle, it can be of great importance for the core. The latitudinal distribution of the CMB heat flux has indeed been shown to play a role in geodynamo simulations (Glatzmaier et al., 1999; Kutzner and Christensen, 2004; Olson et al., 2010). TPW has been suggested by several authors as a source of time variation for the geodynamo, notably concerning the reversal frequency of the magnetic dipole (Courtillot and Besse, 1987; Zhang and Zhong, 2011; Biggin et al., 2012). These fast TPW-induced time variations could thus be related to abrupt changes of the magnetic dipole behaviour such as the emergence or the end of superchrons. It is all the more important to consider these variations that they can occur on timescales potentially faster than typical variations originating solely from mantle convection. As well as short timescales, the CMB heat flux outputs in the different cases includes small length scales. Those small scales can be seen in the heat flux snapshots of Fig. 2 and Fig. 3, and they also appear in the patterns of the first PCA components in Fig. 11 and Fig. 12. Attempts to relate geodynamo models with mantle convection have previously been made using either degree 1 or 2 fixed heat flux patterns, or tomographic patterns derived from seismic tomography (Olson et al., 2010; Terra-Nova et al., 2019; Mound and Davies, 2023). These attempts either overlook the effect of small scales in the CMB heat flux, or focus on a present-day heat flux pattern which may not be representative of the history of Earth's mantle convection.

## 4.2 Effect of chemical piles on the CMB heat flux

The positions of chemical piles and subducted slabs in the lower mantle dominates the large scale CMB heat flux signal of both MF and MC models. Piles keep the mantle material warm below them, while subducted slabs cool the surrounding mantle. Piles can also be correlated to large-scales geoid anomalies and be affected by TPW. The behaviour of the combined system formed by piles and slabs is thus of prime importance regarding the CMB heat flux and the way it is redistributed by TPW. It has been suggested by studying the locations of plume generation zones in the lower mantle that the chemical piles have stayed

fixed over at least 300 Myr (Burke et al., 2008; Torsvik et al., 2010; Dziewonski et al., 2010). Such stability requires that the piles are not significantly altered by mantle convection, and that they are associated with a long-term positive geoid anomaly. If this is indeed the case, TPW only occurs around an axis which passes through the two antipodal piles (Dziewonski et al., 2010; Torsvik et al., 2014). In this case, the positions of the piles relative to the equator would not be affected, and the TPW would not significantly modify the large-scale CMB heat flux pattern. Instead, the CMB heat flux would be dominated by a strong degree 2 order 2 component. As it can be seen in Fig. 8 and Fig. 9, the degree 2 order 2 component of the CMB heat flux does not dominate during the whole duration of the simulation in any of the cases. The $z_{2,2}$ coefficient however dominates for most of the simulation in cases MF2 and MF*. In cases MF0 and MF*, the degree 2 order 2 pattern largely dominates for the last 240 Myr of the model. This is consistent with the expectations of stable piles at the equator. The degree 2 order 2 pattern slowly fades out when coming back in time in both cases, as it is transferred to a degree 2 order 0 pattern. A strong degree 2 order 2 pattern however reappears at -440 Myr in MF*. The piles play also a key role in the distribution in latitude of the CMB heat flux. In case MC1, the negative geoid anomalies above chemical piles impose the piles to move towards the poles. In Fig. 9, this results in a negative degree 2 order 0 spherical harmonic coefficient of the CMB heat flux (large equatorial heat flux) throughout the whole simulation. In case MF2, the geoid anomaly above chemical piles is mostly positive, forcing the piles to stay at low latitudes while polar regions are cooled by subducting slabs. In Fig. 8, the degree 2 order 0 coefficient of the CMB heat flux thus stays positive (low equatorial heat flux). In case MF1, for which the correlation between the geoid and chemical piles is weaker, the $z_{2,0}$ coefficient changes sign during the simulation. This is also the case in MF*, for which the correction does not depend on the position of the piles.

### 4.3 PCA of the CMB heat flux

The different PCA we computed give the dominant heat flux patterns for each case. The PCA components of the CMB heat flux largely reflects the positions of the piles. The insulating effect of the piles indeed dominates the large scales of the CMB heat flux. The dominant length scales of the heat flux patterns tend to decrease with the increase of the PCA component number, but the length scale separation between the three first components is rather weak. It is particularly true for the TPW corrected cases. This is due to the addition of a source of complexity in these cases, which results in more different large scales components required to explain the CMB heat flux time series. This additional source of complexity in the dataset also translates in a smaller amount of variance explained by the first PCA components, as shown in Fig. 10.

### 4.4 Limitations and perspectives

#### 4.4.1 Geoid scattering in plate-driven models

The strong instabilities of the "Total geoid" in model MF due to the imposed changes in surface conditions is an important drawback of this geoid. These instabilities translate into fast motions of the poles. Though the rotation rate in case MF1 using this "Total geoid" lies within expected values for the Earth, the pole wanders significantly faster than in the other cases that are not affected by this problem. The pole instabilities create high frequency variations of the CMB heat flux throughout the

whole simulation that are probably unrealistic. Our attempt to remove the surface contribution to the geoid by computing the "No LVVs geoid" is successful at cancelling the high frequency variations in the geoid. The geoid pattern is however strongly impacted. The anomalies associated with subduction zones and chemical piles are notably affected, cancelling the first one and changing the sign of the second. This implies a widely different TPW path in case MF2 compared to case MF1. This path largely differs from the displacement of the magnetic dipole in both cases, which shows the inconsistency between both the "Total geoid" and the "No LVVs geoid" with the paleomagnetic reference frame in our plate-driven mantle convection model. A perspective would be to eliminate spurious fast variations of the geoid while securing an agreement between the inertia axis of the mantle circulation model and the Earth's spin axis, as given by the magnetic dipole axis, or better by palaeogeographic constraints. Such constraints could be implemented in a data assimilation scheme (Bunge et al., 2003; Bocher et al., 2016).

### 4.4.2   Reference frame of the plate reconstruction

Plate-driven models introduce another complexity due to the choice of reference frame in which the plate reconstruction is considered. Depending on the reference frame, the surface can undergo net rotations, as it is the case in the plate reconstruction of Merdith et al. (2021) given in a paleomagnetic reference frame. The correct way to handle net rotations of the surface in plate-driven mantle convection models is still in debate. Some of these net rotations are due to differential rotations between the lithosphere and the mantle while an other part of the net rotations are due to a solid-body rotation of the whole mantle (Rudolph and Zhong, 2014; Coltice et al., 2017). Our model MF is driven by a plate reconstruction in a no-net-rotation reference frame. A different choice could have been made, as it is unclear that this reference frame is the most geophysically appropriate (Müller et al., 2022). For case MF*, we chose to rotate the output of model MF in the paleomagnetic reference frame, by simply applying the net rotations of the surface in the paleomagnetic reference frame to the whole mantle. This method implies potentially unrealistically large lateral displacement of the deep mantle, and assumes that all the displacements of the paleomagnetic pole relative to continents are due to a solid-body rotation of the mantle. Another possibility would be to directly drive the convection model with the plate reconstruction in the paleomagnetic reference frame, as done by Dannberg et al. (2024), with the drawback that it forces spurious flow in the deep mantle. We do not consider imposing a no-net rotation at the CMB as in Steinberger et al. (2019a) as an appropriate alternative. Note however that this indetermination does not exist in self-consistent mantle convection models.

### 4.4.3   PCA patterns and rotation in longitude

From the point of view of core dynamics, rotations in longitude of a given CMB heat flux pattern are irrelevant. The PCA computed here would however see two identical patterns rotated in longitude as two distinct components. The analysis conducted in this study could thus be improved by gathering patterns that are similar by rotations in longitude into the same component.

## 5 Conclusion

The two main goals of this work are to investigate how TPW can affect the CMB heat flux in term of space and time behaviour, and to provide heat flux maps representative of ∼1 Gyr of mantle evolution in a reference frame useful for geodynamo models. Notwithstanding current limitations in predicting the geoid from forward mantle convection models, we show that TPW can greatly affect the large scales of the CMB heat flux by changing the hierarchy between low-degree spherical harmonics modes, by changing the sign of these modes, and by adding time variations on time scales shorter than often considered as realistic. We performed a principal component analysis to obtain the dominant heat flux patterns at the CMB in the different considered cases. These patterns represent the long timescale behaviour, and they preserve small length scales that could be important for dynamo action. These patterns can thus be used as an alternative to snapshots to study implications of mantle convection models on geodynamo models.

This work also highlights the need to better constrain long-term mantle convection models using the moment of inertia of the mantle. Aiming for core-mantle coupling, self-consistent mantle convection models have to be repositioned in a frame that keeps the maximum inertia axis aligned with the rotation axis. This correction can also be used in plate-driven models. In this case, however, it is also possible to rotate the mantle convection model in a paleomagnetic reference frame if the relation between the paleomagnetic reference frame and the mantle reference frame of the model is known. This second option is not consistent with the model itself, and does not ensure that the maximum inertia axis is aligned with the rotation axis. In this case, the potential differential rotation between the lithosphere and the deep mantle is poorly constrained. The next generation of models driven by reconstructed plate motions could consider including consistency with true polar wander as a constraint for data assimilation.

*Code and data availability.* Full data files in HDF5 formats are available for the models outputs and CMB heat flux PCA results, as well as movies of the outputs an python scripts on Zenodo (Frasson et al., 2024, *https://doi.org/10.5281/zenodo.10523321*).

*Author contributions.* HCN and SL conceived the project, NC and NF provided data of mantle numerical simulations and computed additional outputs from the original models, TF wrote and ran all analysis scripts, prepared all figures, and led the writing of the manuscript. All authors discussed the results and contributed to the writing of the final manuscript.

*Competing interests.* The authors declare no competing interest.

*Acknowledgements.* The SHTns library (Schaeffer, 2013) was used for spherical harmonics transforms and rotations. We thank Nathanaël Schaeffer for implementing spherical harmonic rotations within SHTns. We thank the three reviewers, and notably Bernhard Steinberger and

Shijie Zhong, for pointing out an error in the original computation of the geoid in model MC. This work was supported by the French Agence

Nationale de la Recherche under grant ANR-19-CE31-0019 (revEarth).

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
