# Peer review of "On the impact of true polar wander on heat flux patterns at the core-mantle boundary"

_EGUsphere, 2023_

## Author Comment (AC5)

**Geoid computations in support of the reply to comments on 'On the impact of true polar wander on heat flux patterns at the core-mantle boundary' submitted to Solid Earth by Thomas Frasson, Stéphane Labrosse, Henri-Claude Nataf, Nicolas Coltice and Nicolas Flament'**

November 24, 2023

[Figure]

Figure 1: a) Corrected geoid newly computed for the snapshot at -250 Myr in model MC. b-f) Test geoids obtained for the same snapshot as a) following different methodologies. No topography: geoid computed without taking the topography into account (i.e. geoid signal produced by the density distribution only). No density: geoid computed without taking the density into account (i.e. geoid signal produced by the topography only). No LVVs: geoid computed after canceling the lateral variations of viscosity and density in the upper 350 km (same as the No LVVs geoid of the MF model in the manuscript). Two layers: the viscosity is taken homogeneous in the upper mantle and in the lower mantle, with an increase of viscosity only at 670 km depth by a factor 30. Radial: geoid computed using the radial viscosity profile shown in Figure 2, corresponding to the laterally-averaged viscosity in model MF at time t = 0 Myr. g) Originally submitted geoid shown in the manuscript. h) Surface topography for the snapshot at -250 Myr in model MC. i) Position of the basal mantle structures (shown in gray) at the CMB for the snapshot at -250 Myr in model MC.

[Figure]

Figure 2: Viscosity profiles in the different geoid tests shown Figure 1. The 'Corrected' profile corresponds to the viscosity associated with the mean temperature profile in model MC. The centers and rims of continents have a viscosity respectively 100 times and 50 times higher than this profile. The basal mantle structures are 10 times more viscous that this profile. The 'Two layers' viscosity profile imposes a viscosity 30 times higher in the upper mantle than in the lower mantle. The 'Radial' profile is the laterally-averaged viscosity in model MF at time t = 0 Myr. On Figure 1, the 'Corrected' profile corresponds to cases a-d, the 'Two layers' profile corresponds to case e, and the 'Radial' profile corresponds to case f.

---

## Author Response (AR1)

**Referee #1**

Page 2: lines 35-45: perhaps discuss a short part about how we think that the chemical (and therefore negatively buoyant) heterogeneity may be confined to a small region at the base of the LLVPs. See Richards et al (2023; EPSL: "Geodynamic, geodetic, and seismic constraints favour deflated and dense-cored LLVPs").

Thank you for pointing out the recent study of Richards et al (2023). We refer to it in section 1:

"Recent works suggest a chemically distinct composition at the base of these structures, stabilizing them by imposing a negative buoyancy (Richards et al., 2023)."

Page 6: lines 160-165: It would be good to elaborate on why there are two distinct ways to compute the geoid. This study will likely attract a varying audience (e.g., core dynamicists), so some background on this -- even just 2-3 sentences -- would be helpful. What does zeroing out the upper 350 km achieve?

We made the purpose of the No LVVs geoid clearer in the manuscript. We notably included these lines in section 2.2:

"The computation of the geoid is very sensitive to large lateral viscosity variations in the mantle (Čadek and Fleitout, 2003; Flament, 2019). The MF model is driven by a plate reconstruction model, updated every 1 Myr, which notably imposes the positions of viscous slabs. The update of the slab positions strongly affects this "Total geoid", creating discontinuities in the time evolution of the geoid. To tackle this issue, we compute a second geoid, called "No LVVs geoid", for which we cancel out density and viscosity lateral variations in the upper 350 km."

"Radial viscosity profiles are classically used to compute the geoid using geoid kernels (Richards and Hager, 1984; Rouby et al., 2010; Steinberger et al., 2019). In our plate-like models, the largest lateral variations of viscosity happen in the upper mantle. Removing the effects of these lateral variations in the upper mantle thus enables us to compute a geoid that is closer to the one computed from radial geoid kernels."

Why is the "No LVV" method not applied to the MC model? If this is because it was not calculated in the original study, is there a way to use the output you have access to to apply the same "No LVV" method. It would be better for comparison. Or maybe this is not applicable? If so, please explain why.

An explanation for the reason why a No LVVs geoid was not used in model MC is given in section 2.2:

"The MC model is fully self-consistent without any forcing at the surface by imposing a free-slip boundary condition. The Total geoid computed in MC evolves smoothly, preventing the scattered TPW observed in MF1. It is thus not necessary to remove the effect of lateral variations in the upper mantle to obtain a smoother geoid as it was the case in the MF model, and only the "Total geoid" is computed for this model"

Page 9: lines 250-260. If the variations in the geoid are so large compared to today's actual geoid, what does this mean in terms of how "earth like" the CMB predictions would be? I realize now that you explain this later in the Discussion, so perhaps point the reader to it.

A reference to the discussion has been added in section 3.2:

"The impacts on the CMB heat flux of the different geoid behaviors are discussed in Sect. 4.3"

Page 19: Section 4.2. Is it possible to give some idea of the timescales of the PCs? Perhaps even estimate the frequency content of the time series. Since these undulations time (figs 10-11) reflect the mobility of the piles, can these be related to subducting slabs from above? Can you potentially derive some timescale for surface events to be translated to CMB events? I think this would be very interesting.

We agree that it would be interesting to link the components with specific events in the mantle convection models. This exercise is however challenging and would have significantly complexified the manuscript. We thus decided not to include discussions about how the components relate to the timescales of mantle convection processes in the revised version.

**Bernhard Steinberger**

My main problem is that the geoid results shown are very different from the real Earth, and also very different from each other. It is not clear to me at all where these differences come from. It is all said in the end of the conclusion: "it would be of great interest to understand where these discrepancies come from". I agree, and I think it should be done in this paper and not in some future work -- among other things, in order to reduce the chance that these discrepancies come from actual errors in the computations.

We indeed discovered an error in the computation of the geoid in model MC. The results and discussion have been updated accordingly in the manuscript. We thank you again for insisting that we clarify this issue.

More specifically, the methods would also have to be better described. It may be possible to extract these from the literature given, but at least some essentials need to be discussed: Particularly, what rheological model is used? Since the geoid strongly depends upon it, in particular on (average) radial viscosity structure. Is it the same in the MF and MC models, or different? If it is the same, why the geoids are so different? Also, the CMB heat flux is different for MF and MC models (line 245); which CMB temperature do they use? Is it the same?

We provide in the revised version the time-averaged profiles of viscosity and temperature in the mantle for both models. A figure showing these profiles has been added to the manuscript in section 3.1.

The two cases MF1 and MF2 start more similar, but then evolve increasingly different. Does the density structure (below 350 km depth) also evolve differently in the two cases, or is it at each time the same density (below 350 km), only the geoid is computed

differently? I think this would make more sense, i.e. you always insert slabs at each time step, but why would differences increase with time then?

We clarified the way the "No LVVs geoid" is computed by adding this precision in section 2.2:

"The density and viscosity distribution in the mantle below 350 km is not modified."

We also mention that the similarity between the two geoids in model MF at the beginning of the simulation might be due to an effect of initial conditions in section 3.2:

"At the beginning of model MF, the piles are associated with positive anomalies in the "Total geoid'", as it can be seen in the snapshot at -900 Myr. This positive signal above the piles only lasts for the first 150 Myr of the simulation. This change of sign in the geoid anomalies above the piles could thus be an effect of the initial conditions."

Also, what boundary condition is used for geoid computation in MF? What I usually do is I use prescribed plate motions only for the flow and advection calculation, but free-slip for the subsequent geoid computation at each time step. Because prescribed surface motions are appropriate for flow computations, but may not give realistic surface radial stresses and topography, hence not realistic geoid. This would be important to know in order to understand the geoid in this case.

We mentioned the boundary condition used in model MF in section 2.2:
"Model MF is forced at the surface by the plate model. To compute the geoid at a given time step, we re-start the model at this time-step with a free-slip condition at the surface as usually done in plate-driven models (Steinberger, 2016; Flament, 2019; Mao and Zhong, 2021)."

Regarding results, why the geoid in case MC has such high amplitude and is such strongly correlated with continents? In reality, continents are mostly isostatically compensated at shallow levels and are associated with a very weak signature, i.e. there is hardly any correlation between geoid and the continent-ocean distribution. I think something is wrong here. On lines 286/287 you write that piles are mostly associated with geoid lows, but I don't see this; I see just the correlation with continents.
The correlation between the geoid and continents no longer exists in the corrected geoid. We modified the results and discussion accordingly.

And why there is no such strong continent signal in MF? The difference in results between the three cases MC, MF1 and MF2 is really puzzling and some analysis should be given to understand the differences, e.g. by separating different contributions (topography, Moho, mantle density down to 350 km, mantle density below 350 km, CMB topography.
The difference between the geoid produced by model MC and MF is much smaller after correction of the geoid in model MC. The differences between the Total geoid in models MF and MC are discussed in section 4.3.2.

**Shijie Zhong**

One class of convection models (MF) presented in this study used CitcomS. Given that CitcomS has been extensively benchmarked for the geoid problems, I would think that the geoid results from this class of models should be okay. However, the geoid results from this class of calculations also raise some concerns to me. For example, I do not quite understand why the geoid would be so different after the removal of shallow thermal structure (i.e., the top 350 km), because the long-wavelength geoid (e.g., at degree-2) often is insensitive to buoyancy at shallow depths where the geoid kernel goes to zero (the geoid kernel concept remains largely relevant even for models with 3D mantle viscosity).

The No LVVs geoid shown in the manuscript is computed by cancelling both the density heterogeneities and the viscosity heterogeneities above 350 km depth. The large differences between the No LVVs geoid and the Total geoid arise from the removal of the lateral viscosity variations rather than from the removal of the lateral density variations. We tried computing a geoid suppressing only the density variations above 350 km depth, showing very few differences with the Total geoid. We added a mention of this third geoid output in the manuscript in section 2.2:
"A third geoid has been computed by cancelling only the lateral variations of density above 350 km depth. The geoid produced in this case is very close to the "Total geoid'and the TPW path does not significantly differ. We thus discarded this case for this study."

Anyway, it is unclear from the manuscript how the geoid and dynamic topography were computed (e.g., were free-slip boundary conditions used together with some appropriate lithospheric viscosity?). The authors need to describe these issues in the revision. Prescribed surface velocity boundary conditions tend to produce spurious pressure field and hence dynamic topography, with high viscosity lithosphere.

We clarified this point in section 2.2:
"Model MF is forced at the surface by the plate model. To compute the geoid at a given time step, we re-start the model at this time-step with a free-slip condition at the surface as usually done in plate-driven models (Steinberger, 2016; Flament, 2019; Mao and Zhong, 2021)."

The other class of convection models (MC) produced geoid anomalies of several kilometers that are only slightly smaller than that of surface dynamic topography (Fig. 2). These calculations are presumably for Rayleigh numbers that are comparable with that for the Earth's mantle and with that in the first class of convection models using CitcomS. In this type of situation, kilometers of geoid anomalies seem too large to me. Additionally, the geoid to topography ratio in most models in general should be around 0.1-0.2 (like admittance which is the ratio of gravity anomalies to topography, the geoid to topography ratio is only sensitive to viscosity structure, but significantly less sensitive to distribution of buoyancy).
We thank you for pointing out this problem. We indeed discovered an error in the computation of the geoid in model MC. After correction, geoid anomalies are much

smaller (while still larger than present-day anomalies), and their pattern is much closer to expectations. We modified the manuscript to discuss the corrected geoid.

**Additional modifications**

Additional modifications not mentioned above have been made to this revised version. These changes are mostly the results of the addition of a new case, called MF* in the new version, as an alternative to the TPW correction performed in the other cases. In this case, the outputs of the MF mantle convection model were rotated in the paleomagnetic reference frame of Merdith et al. (2021). The addition of this alternative case was motivated by the recently submitted work by Dannberg et al. (2023), in which a mantle convection model is driven by the plate reconstruction of Merdith et al. (2021) kept in the original paleomagnetic reference frame. Our MF model is driven by the same plate reconstruction but in a no-net-rotation reference frame in order to avoid net rotations of the lithosphere. We simply rotated the outputs of model MF in the original paleomagnetic reference frame by considering that the net rotations of the surface in the plate reconstruction are rotations of the whole mantle. This is overall similar to running the mantle convection model in the paleomagnetic reference frame and applying the net rotations of the surface to the whole mantle, as done in Dannberg et al. (2023). We used this case to compare the evolution of the CMB heat flux in the paleomagnetic reference frame with the TPW corrected cases, and we also added a comparison between the positions of the maximum inertia axis and the magnetic dipole axis in the paleomagnetic reference frame of Merdith et al. (2021).

Other minor corrections have also been done in the text and the figures, including:
- Change of colormap for the CMB heat flux and geoid maps (Fig. 2, Fig. 3, Fig. 11 and Fig. 12).
- Change of the color scale of the geoid in Fig. 2 and Fig. 3 to avoid the saturation.
- Addition of the time-averaged spectra of the CMB heat flux in models MF and MC (FIg. 4).
- Correction of the latitudes in Fig. 11: the heat flux patterns were upside down due to a wrong sign in the definition of the latitude for this figure.

**References**

Dannberg, J., Gassmoeller, R., Thallner, D., LaCombe, F., and Sprain, C.: Changes in core-mantle boundary heat flux patterns throughout the supercontinent cycle, arXiv preprint arXiv:2310.03229, 2023.

Merdith, A. S., Williams, S. E., Collins, A. S., Tetley, M. G., Mulder, J. A., Blades, M. L., Young, A., Armistead, S. E., Cannon, J., Zahirovic, S., et al.: Extending full-plate tectonic models into deep time: Linking the Neoproterozoic and the Phanerozoic, Earth-Science Reviews, 214, 103 477, 2021.

---

## Author Response (AR3)

**Bernhard Steinberger**

One addition is the model MF\*. From what I understand, in this model the positions of plates and boundaries are rotated into the paleomagnetic reference frame, but rotation rates are additionally modified to change them into a no-net-rotation reference frame. But these additonal rotation rates are not integrated to change the positions of the plate. At least, I think this would be the most sensible way of doing things, because it would maintain the paleomagnetic reference frame, yet lithospheric net rotations would not cause a net rotation of the entire mantle. Yet it is not entirely clear that this is your procedure, so I think you should make it explicit. A few more points in this context:

The MF\* alternative case is not a new convection model, it is simply the MF model rotated in the paleomagnetic reference frame. In this reference frame, the lithosphere undergoes net rotations. Because we simply rotate the outputs in the paleomagnetic reference frame, the net rotations of the lithosphere also apply to the underlying mantle. We explained our procedure more clearly in section 2.3 lines 242-266 (in the trackchanges version).

The net rotations of the deep mantle in the MF\* case can be seen in the supplementary video *qcmb_MF.mp4*. A significant eastward rotation notably occurs between -700 Myr and -580 Myr. This net rotation is due to the coupling with the surface net rotation. Similar rotations of the deep mantle are visible in the supplementary videos to Dannberg et al. (2024) (such as the eastward drift between -700 Myr and -580 Myr). Though our approach is different from the one of Dannberg et al. (2024), the strong coupling between the lithosphere and the deep mantle in their models seems to produce very similar net rotations of the deep mantle.

Around line 92: One should mention that net rotation can also be a problem in mantle reference frames: Plate reconstructions in a (e.g. hotspot-based) mantle reference frame do contain a net rotation, but when these plate reconstructions are imposed as boundary conditions, they will usually not yield the same net rotation relative to the deep mantle - e.g. in the case without lateral viscosity variations, the deep mantle will show the same net rotation, and there is no relative net rotation. So even in the case of the mantle reference frame, it would probably be better to remove net rotation from the rotation rates (but without modifying the positions of plates and boundaries by integrating rotation rates). As an alternative procedure, I change from free-slip to fixed CMB at degree-1 toroidal (the net rotation component) flow, in my computations, in this way substantially reducing the net rotation of the deep mantle and introducing a net rotation of lithosphere vs deep mantle even without lateral viscosity variations.

We are aware of this potential net rotation of the deep mantle when the plate reconstruction model itself contains net rotations. We agree that it would be interesting to run an end-member MF model in which net rotation would be removed from the plate

reconstruction and at depth, because net rotation depends on lateral viscosity variations that are limited in model MF. In our MF* case, all the surface net rotations are instead applied to the deep mantle. Some of these net rotations are expected, while some shouldn't exist (if they are differential rotation of the lithosphere relative to the deep mantle). We have stated in the text that the rigid rotations that arise in model MF are a combination of the imposed boundary conditions and considered lateral viscosity variations (lines 249-259).

Line 104/105: "This alternative correction is equivalent to what seems to be done": I think it would be beneficial to figure out how Dannberg et al. (2023) handle this net rotation issue, i.e. if they remove net rotation the same way you do it. If they keep the large amount of net rotation, which is then transferred to the mantle, then this would indeed be somewhat problematic in their computation.

Following the comment by Juliane Dannberg, we explained more clearly the difference between their approach and ours in the manuscript (lines 242-266).

Also, I find lines 124-126 still a bit confusing. From what I understand, they start with a paleomagnetic reference frame, which they convert, at least since 500 Ma, into a "TPW corrected mantle reference frame" (see their Table 1), yet this is not (at least not necessarily) a no-net rotation reference frame. So, conversion into a no-net rotation reference frame is something you would still need to do (not sure whether you do this). What is further confusing is that on line 322 you write that your reconstruction is identical to the "no net rotation" plate reconstruction of Müller et al. (2022). As said, I think it would be most suitable to change rotation rates to a no-net rotation frame, but not to rotate plate positions into it, i.e. keep them in a mantle frame.

The net rotations are indeed removed from the plate reconstruction model, which is identical to the NNR case of Müller et al. (2022). This plate reconstruction corresponds to the one of Merdith et al. (2021) from which all the net rotation has been removed. The plate reconstruction is thus given in a no-net-rotation reference frame which assumes no net rotations of the lithosphere relative to the underlying mantle. We changed the lines you referred to (lines 113-116) by making it clearer that we use the plate reconstruction of Merdith et al. (2021) rotated in the no-net-rotation mantle reference frame as in Müller et al. (2022).

Lines 261 ff: Again (see above comment on lines 124-126) I am not sure whether you are doing (or describing) this right. Merdith et al. use at least since 500 Ma a "TPW corrected mantle reference frame", so in order to get the reconstruction into a paleomagnetic reference frame, one would presumably just have to undo their TPW correction, but that presumably does not remove all net rotation. So, after doing the TPW correction, do you additionally remove any remaining net rotation? But if you do that also

for the cumulative rotations, then you would no longer be in the paleomagnetic reference frame. Sorry, I am still confused.

See lines 242-266 for the description on how the MF* case is obtained.

Lines 13/14: "The average TPW rates ..." - I think it should be clarified that this sentence refers to the alternative TPW correction according to Fig. 7, not to your model.

The average TPW rates we mention here are the ones obtained for cases MF1, MF2 and MC1. We mentioned average TPW rates ranging from 0.4°/Myr to 1.8°/Myr, with MF2 being the lower value and MF1 being the higher value.

Line 206: "densities and flow velocities are first set equal to zero" - are you sure you are setting flow velocities equal to zero? Because density variations below 300 km will also drive flow above 300 km, and setting flow above 300 km to zero would presumably introduce a flow discontinuity which I don't think you want.

This sentence was indeed misleading. As we use a Stokes flow solver to compute the geoid, the velocities in the mantle only depend on the density. There is thus no need to set the flow velocities equal to zero above 350 km. We removed this sentence from the manuscript.

Table 1: I don't understand the sentence "The value of delta rho_c ... is an average over all continents". Do you mean, you use different densities for different continents? Or you use a constant value which is the value obtained by averaging observationally-derived densities for all continents?

Different types of continents are used, with different densities (see Flament et al., 2014). We added a reference to Flament et al. (2014) in the caption of the figure (the reference was already given in the text line 120).

Your new Fig. 1: I don't really understand why the temperature profiles and CMB temperatures are so different (differing by ~1000 K in the lowermost mantle) in the two models. Is that because your model MC does not include adiabatic compression and heating, whereas model MF does? Although the temperature drop in the the bottom thermal boundary layer looks more or less realistic in both cases, I think absolute temperatures are way too low in the model MC. I think this should be explained to avoid confusion.

MC uses the Boussinesq formulation, while MF uses the extended Boussinesq formulation, in which the adiabatic compression effects are accounted for. Those precisions have been added to the description of the models lines 111-112 and line 140.

Line 387: Perhaps "diverge back in time"?

This has been modified

Line 432: Why limited to 9° to 18°? In figure 4 right, it looks like the curve exceeds 20°/Myr.

The TPW velocities are limited between 9°/Myr and 18°/ Myr in the MC1 case because of the time step between two successive geoid snapshots. Maybe you refer to the figure 4 of the initial submission version, in which the geoid was incorrect and the TPW velocity indeed reached 20° /Myr.

Minor comments:
line 82: typo, should be "inertia"
line 271: typo, should be "displacements"
line 410: "as it implies"
A number of references have https://doi.org/ twice

This has been modified

Your response #3:
I did not find the sentence "The density and viscosity distribution in the mantle below 350 km is not modified." in your text.

This was an oversight, it has been added lines 195-196.

**Shijie Zhong**

I still have some concern on MF model's geoid, i.e., the difference between "No LVVs geoid" and "Total geoid", as I pointed out in my first review. The authors explain the difference as a result of lateral variation in viscosity due to slabs. While this may indeed be what is going on, my experience with this sort of model calculations still makes me concerned. Presumably, "No LLVs geoid" calculation should have converged easier because of the removal of lateral variation in viscosity. The effect of lateral variations in viscosity on the geoid is an old topic, and our recent paper [Mao and Zhong, 2021], using the plate motion history for the last 130 Ma and similar temperature- and depth-dependent viscosity to MF models, reproduced the observed geoid from degrees 4 to 12 reasonably well. Admittedly, MF models include much longer plate motion history, and the results could be quite different. I do not want to further delay the publication of this paper, knowing that their calculations prove their main idea.

Viscosity variations are only removed in the top 350 km for the 'No LVVs geoid', which generally converge faster, and occasionally do not converge. We note that the viscosity of the asthenosphere in model MF is not as low as that preferred by Mao and Zhong (2021), which could partly explain the difference in geoid calculation.

line 190, "Viscosity lateral variations are also removed above 350 km …". Can the authors clarify how they did this? What is the viscosity used for the top 350 km then? Is it the horizontally averaged viscosity (like in Fig. 1) for the top 350 km used here?

The 'No LVVs' geoids were obtained by solving the instantaneous Stokes flow in a CitcomS restart from 'velo' files in which the temperature was set equal to ambient mantle temperature above 350 km depth.

Equation 1 for the geoid anomalies. First, l=1 should not be included in this equation, because by definition, there can not be degree 1 gravity or geoid anomalies. Geoid and gravity anomalies start with degree 2. Second, the geoid spherical harmonic expansion coefficients c and s in this equation are dimensionless, and they are scaled by radius of the Earth R to get the geoid anomalies. However, this equation is a bit strange to me (not necessarily incorrect, because one can scale the geoid in anyway). Perhaps, the authors should double check on it.

Equation 1 has been corrected to start the sum at l=2. The same equation with the same scaling was used in Phillips et al. (2009), Greff-Lefftz and Besse (2014) or Rouby et al. (2010).

MF* case with a paleomagnetic reference frame is a good addition. However, given that degree-1 toroidal plate motion (or net rotation of lithospheric shell) is present in present-day plate motion in hotspot reference frame, and that it can be dynamically generated using proper plate boundary viscosity [see Mao and Zhong, 2021], the reference frame seems always an open question. Perhaps the so-called geologically inferred TPW is actually not TPW. I am not sure what I expect the authors to revise on this point, rather than to remind them the complexity of this sort of issues.

We are aware of the complexity of this subject. For our MF* case, we chose to simply rotate the outputs of the MF model in the paleomagnetic reference frame. By doing so, we imposed that the net rotations of the deep mantle were a combination of the net rotation of the deep mantle in the MF model and the net rotation of the surface in the paleomagnetic reference frame. Though this is clearly not an ideal solution, we chose this approach as it only required rotations of the MF model outputs. As you mention, it is difficult to decipher the amount of the surface net rotations due to a solid-body rotation of the mantle. In our approach, we consider that all the surface net-rotations are due to a solid-body rotation of the mantle. A discussion of this problem has been added (section 4.4.2).

**Reply to Juliane Dannberg's public justification**

Dear Juliane,

Thank you for the explanations you provide in your comment, which clarify the differences between your approach and ours. In addition to the differences between the mantle convection models used in our study and in Dannberg et al. (2024), our MF* case differs by the coupling between the net rotations of the surface and the deep mantle. In our model, all the net rotations of the surface are solid rotation of the whole mantle-lithosphere system, while the coupling with the deep mantle is part of the model in Dannberg et al. (2024).

You mention that the differential rotation between the lithosphere and the mantle is probably underestimated in the models you use. The net rotations of the deep mantle relative to the surface can thus be expected to be relatively similar between the two approaches. Our MF* case assumes that all the net rotations of the lithosphere in the plate reconstruction of Merdith et al. (2021) are due to a solid-body rotation of the mantle. Though this is indeed what is expected in the case of TPW, it is less clear whether rotations in longitude should be treated this way. Though the approach in Dannberg et al. (2024) has the advantage of being consistent with the tectonic reconstruction in the paleomagnetic frame of reference, it requires running a specific model, and we instead chose to use our approach which only requires rotations of the outputs of model MF.

We agree with your statement that the net rotation of the mantle does not affect the power of different spherical harmonic degrees of CMB heat flux, nor your q* parameter. It does however play a role in the latitudinal heat flux distribution (your Figure 9b), which is found to strongly affect the geodynamo behaviour (Olson et al, 2010; Zhang and Zhong, 2011).

Best regards,

Thomas Frasson, Stéphane Labrosse, Henri-Claude Nataf, Nicolas Coltice and Nicolas Flament

**References**

Dannberg, J., Gassmoeller, R., Thallner, D., LaCombe, F., & Sprain, C. (2024). Changes in core-mantle boundary heat flux patterns throughout the supercontinent cycle. Geophysical Journal International, ggae075.

Müller, R. D., Flament, N., Cannon, J., Tetley, M. G., Williams, S. E., Cao, X., ... & Merdith, A. (2022). A tectonic-rules-based mantle reference frame since 1 billion years ago–implications for supercontinent cycles and plate–mantle system evolution. Solid Earth, 13(7), 1127-1159.

Merdith, A. S., Williams, S. E., Collins, A. S., Tetley, M. G., Mulder, J. A., Blades, M. L., ... & Müller, R. D. (2021). Extending full-plate tectonic models into deep time: Linking the Neoproterozoic and the Phanerozoic. Earth-Science Reviews, 214, 103477.

Flament, N., Gurnis, M., Williams, S., Seton, M., Skogseid, J., Heine, C., & Müller, R. D. (2014). Topographic asymmetry of the South Atlantic from global models of mantle flow and lithospheric stretching. *Earth and Planetary Science Letters*, *387*, 107-119.

Mao, W., & Zhong, S. (2021). Constraints on mantle viscosity from intermediate‐wavelength geoid anomalies in mantle convection models with plate motion history. *Journal of Geophysical Research: Solid Earth*, *126*(4), e2020JB021561.

Phillips, B. R., Bunge, H. P., & Schaber, K. (2009). True polar wander in mantle convection models with multiple, mobile continents. *Gondwana Research*, *15*(3-4), 288-296.

Greff‐Lefftz, M., & Besse, J. (2014). Sensitivity experiments on True Polar Wander. *Geochemistry, Geophysics, Geosystems*, *15*(12), 4599-4616.

Rouby, H., Greff-Lefftz, M., & Besse, J. (2010). Mantle dynamics, geoid, inertia and TPW since 120 Myr. *Earth and Planetary Science Letters*, *292*(3-4), 301-311.

Olson, P. L., Coe, R. S., Driscoll, P. E., Glatzmaier, G. A., & Roberts, P. H. (2010). Geodynamo reversal frequency and heterogeneous core–mantle boundary heat flow. Physics of the Earth and Planetary Interiors, 180(1-2), 66-79.

Zhang, N., & Zhong, S. (2011). Heat fluxes at the Earth's surface and core–mantle boundary since Pangea formation and their implications for the geomagnetic superchrons. Earth and Planetary Science Letters, 306(3-4), 205-216.